# TACKLING THE XAI DISAGREEMENT PROBLEM WITH ADAPTIVE FEATURE GROUPING

**Gabriel Laberge**[*]**& Ola Ahmad**
Thales cortAIx Lab
Montreal, Qc, Canada
{gabriel.laberge,ola.ahmad}@thalesgroup.com

## ABSTRACT

Post-hoc explanations aim at understanding which input features (or groups thereof) are the most impactful toward certain model decisions. Many such methods have been proposed (ArchAttribute, Occlusion, SHAP, RISE, LIME, Integrated Gradient) and it is hard for practitioners to understand the differences between them. Even worse, faithfulness metrics, often used to quantitatively compare explanation methods, also exhibit inconsistencies. To address these issues, recent work has unified explanation methods through the lens of Functional Decomposition. We extend such work to scenarios where input features are partitioned into groups (e.g. pixel patches) and prove that disagreements between explanation methods and faithfulness metrics are caused by between-group interactions. Crucially, getting rid of between-group interactions leads to a single explanation that is optimal according to all faithfulness metrics. We finally show how to reduce the disagreements by grouping features on tabular/image data.

## 1 INTRODUCTION

With the rise in complexity of Machine Learning models, there has also been a rise in concerns regarding the black-box nature of the most performant models. As a result, the field of eXplainable Artificial Intelligence (XAI) has rapidly grown and now proposes a myriad of techniques to "explain" model predictions (Molnar, 2025).

One of the main roadblocks to XAI is the so-called *Disagreement Problem* (DP) (Krishna et al., 2022), which refers to inconsistencies between explanation methods. Due to lack of ground truth in XAI, practitioners cannot decide which explanation, if any, is the correct one when they disagree. To address this issue, methods like Shapley Values (Lundberg & Lee, 2017) and the Integrated Gradient (Sundararajan et al., 2017) have been motivated as the *unique* explanations satisfying a set of theoretical properties. As such, they are advertised as a form of ground-truth. Still, in the case of Shapley Values, it was demonstrated that their "Dummy" property can be violated in practice (Sundararajan & Najmi, 2020). Regarding the Integrated Gradient, its properties were proven to be insufficient at specifying a unique explanation (Lerma & Lucas, 2021).

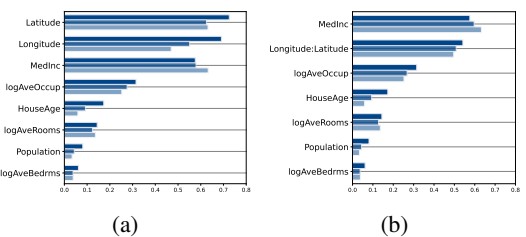

Figure 1: California Housing. (a) PFI, SHAP, and PDP disagree on the importance of `Longitude`. (b) By grouping features, we increase agreement between the techniques.

Alternatively, some works benchmark explainability methods using *(un)faithfulness metrics* : for example the F-score (Tomsett et al., 2020), $\mu$-Fidelity (Bhatt et al., 2020), INFD (Dai et al., 2022), and Shapley-Weighted Fidelity (SWF) (Muschalik et al., 2025). Unfortunately, (un)faithfulness metrics were previously shown to be inconsistent : an explanation can be ranked first by a metric and ranked last by another (Tomsett et al., 2020).

---

[*]Main Correspondance

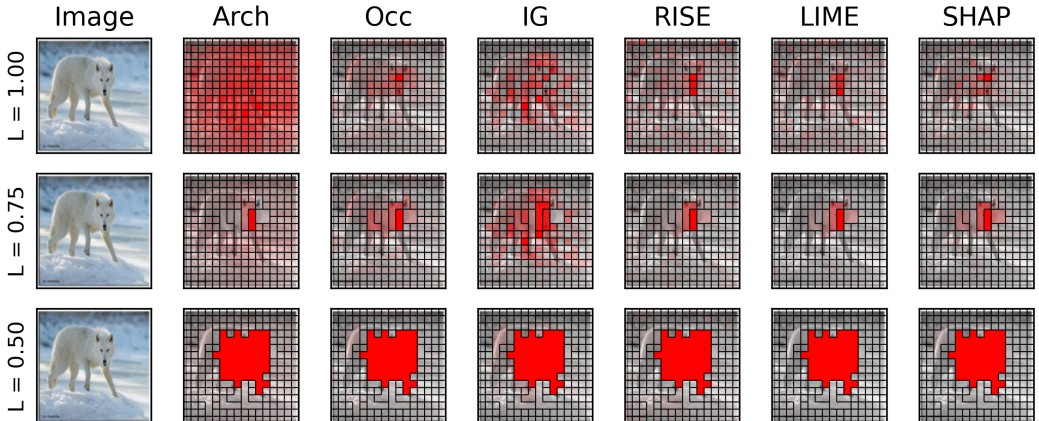

Figure 2: Explaining the "White Wolf" prediction of a ResNet18 using several saliency map methods. (Top), When considering $14 \times 14$ patches, the saliency maps highlight different parts of the image. (Middle) To minimize a disagreement objective function, AGREED fuses the various patches. For example, the initial patches covering the wolf's eyes and nose are fused. (Bottom), Eventually, AGREED leads to large patches where agreement among saliency map techniques is increased.

Recent work has unified the various explanation techniques through the lens of Functional Decomposition (Fumagalli et al., 2025; Deng et al., 2024). We extend these efforts to scenarios where features are partitioned into groups (*e.g.* pixel patches for images), we identify the root cause of disagreements between explainers and (un)faithfulness metrics: *between-group interactions*, and we minimize it by adaptively grouping features. Our framework, called Adaptive Grouping to REduce Explanation Disagreements (AGREED), is empirically assessed on tabular and image datasets. Figures 1 and 2 illustrate AGREED on the California and ImageNet datasets. To resume, our contributions are

1. Unifying group importance methods and (un)faithfulness metrics through Functional Decomposition and demonstrating that disagreements are caused by between-group interactions.

2. Proposing the AGREED algorithm to discover feature groups and assessing its performance on tabular and image datasets.

## 2 BACKGROUND

### 2.1 FUNCTIONAL DECOMPOSITIONS

All notation used throughout the paper is enumerated in Appendix A.1. We let $[d] := \{1, \ldots, d\}$ be a set of $d$ features, $\mathcal{X} \subseteq \mathbb{R}^d$ be the input domain, $\boldsymbol{x} \in \mathcal{X}$ be an arbitrary input, $f : \mathcal{X} \to \mathbb{R}$ be a model, and $\mathcal{D}$ be the data distribution of inputs ($\boldsymbol{x} \sim \mathcal{D}$). Given a feature subset $u \subseteq [d]$, we denote its cardinality by $|u|$. Functional Decomposition aims to represent $f$ as a sum of $2^d$ sub-functions

$$f(\boldsymbol{x}) = \sum_{u \subseteq [d]} f_u(\boldsymbol{x}), \tag{1}$$

where $f_u$ only depends on $(x_j)_{j \in u}$. The term $f_\emptyset$ is a constant, the terms $f_u$ for $|u| = 1$ are called *main-effect* while the terms $|u| \geq 2$ are referred to as $|u|$-*way interactions*. Functional Decompositions are not unique and their definition is often based on a heuristic for removing feature $x_j$. One heuristic consists of freezing $x_j$ at a baseline value $b_j$ using the *replace-function* $\boldsymbol{r}_u : \mathcal{X} \times \mathcal{X} \to \mathcal{X}$ defined as

$$r_u(\boldsymbol{b}, \boldsymbol{x})_j = x_j \text{ if } j \in u \text{ otherwise } b_j. \tag{2}$$

Treating the baseline $\boldsymbol{b} \sim \mathcal{B}$ as random leads to the *Marginal Decomposition* (Fumagalli et al., 2025).

**Definition 2.1** (Marginal Decomposition). *Given a distribution $\mathcal{B}$, the Marginal Decomposition is*

$$f_{u,\mathcal{B}}(\boldsymbol{x}) := \mathbb{E}_{\boldsymbol{b}\sim\mathcal{B}}[f(\boldsymbol{r}_u(\boldsymbol{b},\boldsymbol{x}))] - \sum_{v\subset u} f_{v,\mathcal{B}}(\boldsymbol{x}). \tag{3}$$

When $\mathcal{B} = \delta_{\boldsymbol{b}}$ is a Dirac delta centered at $\boldsymbol{b}$, the decomposition falls back to the so-called Anchored Decomposition (Kuo et al., 2010). If $\mathcal{B} = \mathcal{B}_{\text{ind}} := \prod_{j=1}^{d} \mathcal{B}_j$ (*i.e.* input features are independent), then it becomes the *ANOVA Decomposition* (Hooker, 2004) (see Appendix A.2). Since the Marginal Decomposition is relative to a distribution $\mathcal{B}$, it cannot explain model prediction $f(\boldsymbol{x})$ in isolation. Rather, the main effects and interactions explain the *Gap* $f(\boldsymbol{x}) - \mathbb{E}_{\boldsymbol{b}\sim\mathcal{B}}[f(\boldsymbol{b})]$ between a specific prediction and the average prediction. For tabular data, it is common to let $\mathcal{B} = \mathcal{D}$ be the data distribution (Lundberg & Lee, 2017), while for images it is common to use a single baseline image $\boldsymbol{b}$ : the average color (Ribeiro et al., 2016) or $\boldsymbol{0}$ (Petsiuk et al., 2018).

## 2.2 EXPLAINING FEATURE GROUPS

Many explainability methods explain the joint effect of feature groups, instead of their individual effects. This is the case for saliency maps that first group pixels into super-pixels (Ribeiro et al., 2016; Tsang et al., 2020) or square patches (Zeiler & Fergus, 2014; Petsiuk et al., 2018). Assuming $d$ features are fed to the model, consider a partition of $[d]$ into $D$ disjoint groups. This partition can be described with a function $\mathcal{P} : [d] \rightarrow [D]$ that associates each feature $j \in [d]$ to its group index $\mathcal{P}(j) \in [D]$. We will employ the mapping $\mathcal{P}(u) := \{\mathcal{P}(j) : j \in u\}$ for $u \subseteq [d]$ to enumerate all groups indices within a $|u|$-way interaction and the inverse map $\mathcal{P}^{-1}(U) := \{j \in [d] : \mathcal{P}(j) \in U\}$ for $U \subseteq [D]$ to list all features that are part of certain groups. Finally, $\mathcal{P}'$ is a super-partition of $\mathcal{P}$ if $\mathcal{P}(j) = \mathcal{P}(k) \Rightarrow \mathcal{P}'(j) = \mathcal{P}'(k)$.

When investigating the effect of feature groups on the gap $f(\boldsymbol{x}) - \mathbb{E}_{\boldsymbol{b}\sim\mathcal{B}}[f(\boldsymbol{b})]$, the ideal scenario is that of a Groupwise Additive Model.

**Definition 2.2** (Groupwise Additive Model (Sivill & Flach, 2023)). *Let $R \subseteq \mathcal{X}$ be a hyperrectangle region. A model $f : \mathcal{X} \rightarrow \mathbb{R}$ is called Groupwise additive in $R$ w.r.t $\mathcal{P}$ if there exists $D$ functions $g_{\mathcal{P}^{-1}(\{i\})}$ that each only depend on features in group $i$ and such that*

$$f(\boldsymbol{x}) = \omega_0 + \sum_{i=1}^{D} g_{\mathcal{P}^{-1}(\{i\})}(\boldsymbol{x}) \quad \forall \boldsymbol{x} \in R. \tag{4}$$

In this ideal scenario, the contribution of group $i$ toward the gap $f(\boldsymbol{x}) - \mathbb{E}_{\boldsymbol{b}\sim\mathcal{B}}[f(\boldsymbol{b})]$ is unambiguous: $g_{\mathcal{P}^{-1}(\{i\})}(\boldsymbol{x}) - \mathbb{E}_{\boldsymbol{b}\sim\mathcal{B}}[g_{\mathcal{P}^{-1}(\{i\})}(\boldsymbol{b})]$. If Equation 4 does not hold however, there is no longer a unique group attribution. This has caused the development of a myriad of post-hoc explainers: $\phi(f, \boldsymbol{x}, \mathcal{B}, \mathcal{P}) \in \mathbb{R}^D$ that estimate the contribution to each group $i \in [D]$. Many methods are conveniently defined in terms of a coalitional game.

**Definition 2.3** (Grouped Coalitional Game). *Define the coalitional game $\nu_{f,\boldsymbol{x},\mathcal{B},\mathcal{P}}$*

$$\nu_{f,\boldsymbol{x},\mathcal{B},\mathcal{P}}(U) := \mathbb{E}_{\boldsymbol{b}\sim\mathcal{B}}[f(\boldsymbol{r}_{\mathcal{P}^{-1}(U)}(\boldsymbol{b},\boldsymbol{x}))] \quad \forall U \subseteq [D]. \tag{5}$$

*that applies the replace-function simultaneously to all features within the same group.*

Various explainability methods can be expressed as a weighted sum of marginal contributions for group $i$

$$\phi_i^\mu(f, \boldsymbol{x}, \mathcal{B}, \mathcal{P}) = \sum_{U \subseteq [D]\setminus\{i\}} \mu(U)[\nu_{f,\boldsymbol{x},\mathcal{B},\mathcal{P}}(U \cup \{i\}) - \nu_{f,\boldsymbol{x},\mathcal{B},\mathcal{P}}(U)], \tag{6}$$

with $\mu(U) \in \mathbb{R}_+$ such that $\sum_{U \subseteq [D]\setminus\{i\}} \mu(U) = 1$. The joint-PDP (Friedman, 2001) and ArchAttribute (Tsang et al., 2020) both employ $\mu(\emptyset) = 1$ and zero otherwise. The joint-PFI (Au et al., 2022) and Patch-Occlusion (Zeiler & Fergus, 2014) both consider $\mu([D] \setminus \{i\}) = 1$ and zero otherwise. Finally, the SHAP (Lundberg & Lee, 2017) uses $\mu(U) = \binom{D-1}{|U|}^{-1}/D$, while RISE (Petsiuk et al., 2018) defines $\mu(U) = 2^{D-1}$.

The Integrated Gradient (IG) (Sundararajan et al., 2017) is an alternative feature importance that is not easily expressed in terms of a coalitional game. Nevertheless, it is naturally extended to feature groups

$$\phi_i^{\text{IG}}(f, \boldsymbol{x}, \mathcal{B}, \mathcal{P}) = \sum_{j \in \mathcal{P}^{-1}(\{i\})} \mathop{\mathbb{E}}_{\substack{\boldsymbol{b} \sim \mathcal{B} \\ t \sim \text{Uniform}(0,1)}} \left[ (x_j - b_j) \frac{\partial f}{\partial x_j} ((1-t)\boldsymbol{b} + t\boldsymbol{x}) \right], \qquad (7)$$

by summing over all features within group $i$ (Tsang et al., 2020). IG, unlike the other methods, is not black-box since it requires access to output gradients instead of model outputs. Recently, Cai & Wunder (2024) have proposed estimates of the IG that relax these assumptions, making it truly black-box.

The calculation of post hoc explanations can require evaluating expectations involving $\mathcal{B}$. Throughout our experiments, all expectations were estimated via Monte Carlo (MC) with $N$ samples.

## 2.3 DISAGREEMENT PROBLEM

Many post-hoc explainers have been proposed, each with different motivations. So, it is not surprising that they often disagree in practice (Krishna et al., 2022). To characterize said disagreements, we use the $L_2$ metric (Laberge et al., 2024)

$$D_{L_2}(\boldsymbol{\phi}, \boldsymbol{\phi}') := \mathop{\mathbb{E}}_{\boldsymbol{x} \sim \mathcal{D}} [\|\boldsymbol{\phi}(f, \boldsymbol{x}, \mathcal{B}, \mathcal{P}) - \boldsymbol{\phi}'(f, \boldsymbol{x}, \mathcal{B}, \mathcal{P})\|^2], \qquad (8)$$

since we will see shortly that this loss has many desirable properties when used as a minimization objective. Other disagreement metrics proposed in the literature, *e.g.* the spearman correlation (Schwarzschild et al., 2023),

$$2\, D_\theta(\boldsymbol{\phi}, \boldsymbol{\phi}') := 1 - \mathop{\mathbb{E}}_{\boldsymbol{x} \sim \mathcal{D}} \left[ \frac{\langle \boldsymbol{\phi}(f, \boldsymbol{x}, \mathcal{B}, \mathcal{P}), \boldsymbol{\phi}'(f, \boldsymbol{x}, \mathcal{B}, \mathcal{P}) \rangle}{\|\boldsymbol{\phi}(f, \boldsymbol{x}, \mathcal{B}, \mathcal{P})\| \, \|\boldsymbol{\phi}'(f, \boldsymbol{x}, \mathcal{B}, \mathcal{P})\|} \right]. \qquad (9)$$

do not share these properties.

The current literature tackles disagreements between explanation methods by benchmarking them using *(un)faithfulness metrics*. Many such metrics take the form

$$\overline{F}(\boldsymbol{\phi}, f, w) = \mathop{\mathbb{E}}_{\boldsymbol{x} \sim \mathcal{D}} \left[ \sum_{U \subseteq [D]} w(U) \bigg( \sum_{i \in U} \phi_i(f, \boldsymbol{x}, \mathcal{B}, \mathcal{P}) - [\nu([D]) - \nu([D] \setminus U)] \bigg)^2 \right] \qquad (10)$$

$$\underline{F}(\boldsymbol{\phi}, f, w) = \mathop{\mathbb{E}}_{\boldsymbol{x} \sim \mathcal{D}} \left[ \sum_{U \subseteq [D]} w(U) \bigg( \sum_{i \in U} \phi_i(f, \boldsymbol{x}, \mathcal{B}, \mathcal{P}) - [\nu(U) - \nu(\emptyset)] \bigg)^2 \right] \qquad (11)$$

using $\nu \equiv \nu_{f, \boldsymbol{x}, \mathcal{B}, \mathcal{P}}$ and a specific weight $w(U) \in \mathbb{R}_+$ for any $U \subseteq [D]$. Metrics that take the form of a weighted squared error include Sensitivity-n (Ancona et al., 2017), INFD (Yeh et al., 2019), $\mu$-Fidelity (Bhatt et al., 2020), and Shapley-Weighted Fidelity (SWF) (Muschalik et al., 2025). We refer to Appendix A.3.1 for the complete definition of each metric. Note that the choice of weights impacts which explanation is considered optimal: minimizing $\underline{F}$ with $w(U) = 2^{-D}$ as done in LIME (Ribeiro et al., 2016) leads to the Banzhaf index (Tsai et al., 2023), minimizing $\underline{F}$ with $w(U) \propto 1/\binom{D-2}{|U|-1}$ leads to SHAP (Lundberg & Lee, 2017). Accordingly, unless we can unanimously agree on which weighting function $w$ is the correct one, (un)faithfulness metrics simply postpone the disagreement problem instead of fixing it.

The unfaithfulness metrics 10 & 11 both assume that features are partitioned into disjoint groups. This contrasts with recent metrics that allow for overlap between feature groups (You et al., 2025). Differences between metrics with/without overlap are discussed in Appendix A.3.2. In this paper, we focus on faithfulness without overlap since the explainability methods studied require a feature partition.

## 3 METHODOLOGY

Because of the aforementioned inconsistencies, practitioners do not have a mean of determining which unfaithfulness metric, and by extent which explanation method, is correct. Each method

employs a different weighting scheme $\mu$ to aggregate marginal contributions (cf. Equation 6) or a different weight $w$ to aggregate errors of additive reconstructions (cf. Equations 10 & 11). Instead of proposing an optimal choice for either $\mu$ or $w$, our next objective will be to lessen the impact of these choices.

**Theorem 3.1.** *The Arch/Occ/LIME/SHAP/RISE attributions can be expressed via the Marginal Decomposition*

$$\phi_i^\mu(\boldsymbol{x}, f, \mathcal{B}, \mathcal{P}) = \sum_{u \subseteq [d]:\{i\}=\mathcal{P}(u)} f_{u,\mathcal{B}}(\boldsymbol{x}) + \sum_{u \subseteq [d]:\{i\}\subsetneq\mathcal{P}(u)} h(|\mathcal{P}(u)|)f_{u,\mathcal{B}}(\boldsymbol{x}), \quad (12)$$

*where $h$ is different for each $\mu$. Different explainability methods disagree on how to redistribute **between-group interactions** ($f_{u,\mathcal{B}}(\boldsymbol{x})$ with $|\mathcal{P}(u)| \geq 2$) among the groups involved.*

**Theorem 3.2.** *Let $R \subseteq \mathcal{X}$ be a hyperrectangle region such that $supp(\mathcal{D}) \subseteq R$ and $supp(\mathcal{B}) \subseteq R$. Let $\mathcal{P}$ be a feature partition. Whenever the model $f$ is groupwise additive in $R$ w.r.t $\mathcal{P}$, (un)faithfulness metrics that follow Equations 10 and 11 are all simultaneously optimized*

$$F(\boldsymbol{\phi}^\mu, f, w) = 0 \quad (13)$$

*for any weight function $w$ and attribution $\phi^\mu$.*

The proofs are presented in Appendix B.2. Both theorems imply that if there were no between-group interactions (*i.e.* $f_{u,\mathcal{B}} = 0$ whenever $|\mathcal{P}(u)| \geq 2$), then **all explanation methods would agree** on the group importance and **all (un)faithfulness metrics would be minimized**. Therefore, the Disagreement Problem can potentially be tackled by searching for a partition $\mathcal{P}$ regarding which $f$ is group-wise additive, or "almost" group-wise additive. This is trivially achieved by considering a single group containing all features, so we must trade-off explanation agreement with group sizes. Similarly to recent work, we frame this search as an optimization w.r.t $\mathcal{P}$ using a special class of loss functions.

**Definition 3.1.** *A partition loss function $\mathcal{L}_f(\mathcal{D}, \mathcal{B}, \mathcal{P}) \in \mathbb{R}_+$ should respect:*

1. *If $f$ is groupwise additive in $R$ w.r.t $\mathcal{P}$, then for any distributions $\mathcal{D}$ and $\mathcal{B}$ such that $supp(\mathcal{D}) \subseteq R$ and $supp(\mathcal{B}) \subseteq R$, $\mathcal{L}_f(\mathcal{D}, \mathcal{B}, \mathcal{P}) = 0$.*

2. *If $f$ is additive w.r.t to group $i$ (i.e. $f(\boldsymbol{x}) = g_{\mathcal{P}^{-1}(\{i\})}(\boldsymbol{x}) + g_{\mathcal{P}^{-1}([D]\setminus\{i\})}(\boldsymbol{x})$), fusing group $i$ with another one does not impact the loss.*

3. *If $\mathcal{P}'$ is a super-partition of $\mathcal{P}$, $\mathcal{L}_f(\mathcal{B}_{ind}, \mathcal{B}_{ind}, \mathcal{P}') \leq \mathcal{L}_f(\mathcal{B}_{ind}, \mathcal{B}_{ind}, \mathcal{P})$.*

These properties ensure that $\mathcal{L}_f$ is a sensible objective to minimize w.r.t $\mathcal{P}$. Property 1 guarantees convergence once the optimal $\mathcal{P}$ is found. Property 2 encourages the minimization algorithm to only fuse groups that interact with some other. Property 3 suggests that, on tabular data, where $\mathcal{B} = \mathcal{D}$, an iterative algorithm monotonically decreases its loss if features are independent.

**Theorem 3.3.** *The $L_2$ disagreements $D_{L_2}(\phi^{Occ}, \phi')$ between Occlusion and the Arch/LIME/RISE/SHAP explainers respect Definition 3.1.*

**Theorem 3.4.** *The $L_2$ disagreements between explanation pairings that do not include Occlusion (e.g. LIME vs SHAP) break property 3 of Definition 3.1. Moreover, using the Pearson correlation disagreement $D_\theta$ breaks property 2.*

The proofs are presented in Appendix B.3. These theorems highlight that only the $L_2$ disagreement metric and certain explanation pairings should be considered when infering feature groups. Our algorithm, AGREED, partitions features using the $L_2$ disagreement between the PDP/ArchAttribute and PFI/Occlusion

$$\mathcal{L}_f^{\text{AGREED}}(\mathcal{D}, \mathcal{B}, \mathcal{P}) := \underset{\boldsymbol{x} \sim \mathcal{D}}{\mathbb{E}} \left[ \sum_{i=1}^D (\phi_i^{\text{PDP/Arch}}(f, \boldsymbol{x}, \mathcal{B}, \mathcal{P}) - \phi_i^{\text{PFI/Occ}}(f, \boldsymbol{x}, \mathcal{B}, \mathcal{P}))^2 \right] = \sum_{i=1}^D \Psi(i) \quad (14)$$

which are the cheapest explanations to compute. Our minimization of equation 14 w.r.t $\mathcal{P}$ follows: 1) start with a granular partition $\{\{1\}, \ldots, \{d\}\}$; 2) select the group $i$ with the highest potential $\Psi(i)$; 3) compute its pairwise interactions with a set of candidate groups $i'$; 4) fuse group $i$ with the

group $i'$ that yields the maximal pairwise interaction; 5) repeat until the objective (cf. Equation 14) falls below $\epsilon$. We refer to Appendix C for the technical details.

Solving $\min_{\mathcal{P}} \mathcal{L}_f^{\text{AGREED}}(\mathcal{D}, \mathcal{B}, \mathcal{P})$ yields a feature partition that is valid over the support of the distributions $\mathcal{D}$ and $\mathcal{B}$ passed as parameters. In the tabular setting, we set $\mathcal{D}$ and $\mathcal{B}$ to the data distribution so solving $\min_{\mathcal{P}} \mathcal{L}_f^{\text{AGREED}}(\mathcal{D}, \mathcal{D}, \mathcal{P})$ leads to a partition $\mathcal{P}$ useful to explain any data point. For image data, we set $\mathcal{D} = \delta_{\boldsymbol{x}}$ and $\mathcal{B} = \delta_{\boldsymbol{b}}$ to Dirac measures over an image of interest and the baseline. Accordingly, solving $\min_{\mathcal{P}} \mathcal{L}_f^{\text{AGREED}}(\delta_{\boldsymbol{x}}, \delta_{\boldsymbol{b}}, \mathcal{P})$ leads to a partition $\mathcal{P}$ that is only valid for this single image $\boldsymbol{x}$. The partitioning algorithm must be run separately on each image.

Regarding scalability, AGREED has complexity $\mathcal{O}(d^2 N^2)$ (see Appendix C), where $N$ is the number of samples used to estimate expectations. Our quantitative experiments on tabular data employed various models, random seeds, and reported explanation disagreements/unfaithfulness metrics at various stages of AGREED. This required subsampling $50 \leq N \leq 100$ instances to ensure reasonable runtimes. Our qualitative examples on tabular data, however, used $N = 1000$ samples since they did not require using multiple models/seeds and logging metrics at each step of the algorithm. For images, $N = 2$ samples are always used (the image and baseline) so the bottleneck is the $\mathcal{O}(d^2)$ factor, which we control by first partitioning images into $W \times W$ patches and subsequently fusing them.

## 4 RELATED WORK

There are multiple existing methods for partitioning input features. The PAIRWISE algorithm advocated by Tsang et al. (2020) groups features in three-steps: 1) computes all $d(d-1)/2$ pairwise interactions between features; 2) retains only the interaction whose strength is above a threshold $\epsilon$ and organizes them in a graph; 3) defines groups as the cliques of said graph. This algorithm was applied to tabular and image data. The RECURSIVE algorithm proposed by Sivill & Flach (2023) was demonstrated to scale as $\mathcal{O}(d \log d)$ on tabular data. Finally, the IGREEDY algorithm introduced by Xu et al. (2024) for tabular data starts from a granular partition $\{\{1\}, \ldots, \{d\}\}$ and progressively fuses pairs of groups until convergence.

Laberge et al. (2024) recently identified feature interactions as the root-cause of disagreement between PDP/SHAP/PFI explanations and minimized them by restricting the baseline distribution $\mathcal{B}$ to rule-based regions. Although promising for tabular data, rule-based regions do not work on pixels. AGREED, in contrast, minimizes disagreements by partitioning features into disjoint groups, a methodology that works for tabular data and images.

Prior work has already unified most explainability methods through Functional Decomposition. (Deng et al., 2024) have previously unified 14 saliency maps while (Fumagalli et al., 2025) developed a categorization of explanations along three axes: conditional-marginal-anchored decompositions, pure-partial-full explanations, and individual-joint-interactions effects. Unlike ours, these existing frameworks do not consider feature partitions. Moreover, while prior frameworks also highlight interactions as the root cause of disagreements, ours is the first to propose a practical methodology to minimize said disagreements.

Contemporary studies propose explaining prediction in images using *overlapping* pixel patches. To extract patch importance, the model architecture is constrained to $f(\boldsymbol{x}) = \sum_{i=1}^{D} f_i(\boldsymbol{x}_{G_i})$, where $G_i \subseteq [d]$ is the $i$th patch of pixel. Pixel groups could be the receptive field of neurons in a CNN (Brendel & Bethge, 2019), or the result of a sparse attention mechanism (You et al., 2025). This differs from AGREED, which is forced to consider disjoint feature groups to remain model-agnostic. As a result, AGREED might be forced to merge many features together, hindering interpretability compared to model-specific methods that allow for overlap.

## 5 EXPERIMENTS

An open-source implementation[1] is provided to reproduce the experiments.

---

[1] https://github.com/thalesgroup/AGREED

## 5.1 TABULAR DATASETS

All tabular experiments were run on a laptop with an 11th Gen Intel(R) Core(TM) i7-11850H CPU, 16 threads, and 32 GiB of RAM.

### 5.1.1 KNOWN GROUND-TRUTHS ON SYNTHETIC DATA

We compared AGREED with IGREEDY, RECURSIVE, and PAIRWISE on synthetic data where the model $f$ is known to be groupwise additive over $\mathcal{X}$ w.r.t some ground-truth partition $\mathcal{P}^{\star}$. The exact and estimated partitions were compared for any discrepancies. We set $\mathcal{B} = \mathcal{D}$ as the data distribution. For various numbers of dimension $d$, MC samples $N$, and five random seeds, we generated random data/models with correlated features (see Appendix D.1). Since an optimal partition exists, we set $\epsilon = 10^{-10}$.

All algorithms obtained the exact ground-truth partition except for IGREEDY. We suspect that this method does not always converge because its termination criterion assumes that the loss monotonically decreases as features are grouped. This can only be guaranteed when features are independent.

To compare the scalability of each algorithm, we compared the number of model calls across different values of $d$ and $N$. According to Figure 3, AGREED and PAIRWISE have a similar runtime, but it is the recursive method that scales best with $d$ and $N$.

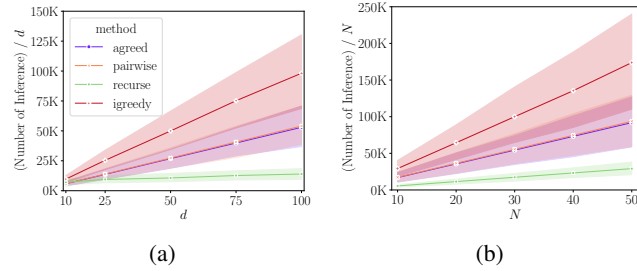

(a)            (b)

Figure 3: Comparing AGREED to baselines on synthetic tabular data. (a) Scalability of each method w.r.t the number of features $d$. (b) Scalability of each method w.r.t the number of Monte Carlo samples $N$. Confidence bands range from $5^{\text{th}}$ percentile to the $95^{\text{th}}$.

### 5.1.2 REAL DATASETS

When studying real-world black-boxes, we no longer know the optimal partitions. We instead evaluate how much can AGREED reduce PDP/SHAP/PFI disagreements and minimize the Sensitivity-1, INFD, and SWF (un)faithfulness scores.

The Marketing[2] ($d = 16$), Default-Credit[3] ($d = 23$), SPAM[4] ($d = 57$), and NOMAO[5] ($d = 118$) datasets were investigated as they contain a large number of features, which renders computing all pairwise interactions intractable. We split each dataset into train/test sets with ratio 0.9:0.1, and we trained Explainable Boosting Machines (EBM) (Nori et al., 2019) from the `InterpretML` library and Histogram Gradient Boosted Trees (HGBT) from the `ScikitLearn` package (Pedregosa et al., 2011). Hyperparameters were tuned using 5-Fold CV and random search.

AGREED requires a random subset of $N$ data samples to approximate expectations w.r.t $\mathcal{D}$. For the Marketing and Default-Credit, we considered $N = 100$ samples and for the other two, we employed $N = 50$ samples. To account for the randomness that arises from subsampling large datasets, we repeated the subsampling of $N$ points five times using different random seeds, ran the partitioning algorithms, and reported the $L_2$ disagreements and Sensitivity-1, INFD, SWF infidelity scores. The runtimes for AGREED ranged from 10 to 100 seconds. Note that the partitioning only needs to be done once and can used to compute local explanation on any data point.

Figure 4 presents the tradeoffs between disagreements/(un)faithfulness and number of feature groups when running AGREED on the Marketing dataset. Other datasets exhibit similar trends, see Appendix D.2.1. From Figure 4 (a), we see that AGREED reduces disagreements between any pairing of explainers although the algorithm is designed to minimize differences between PDP and PFI. This

---

[2] https://archive.ics.uci.edu/dataset/222/bank+marketing

[3] https://archive.ics.uci.edu/dataset/350/default+of+credit+card+clients

[4] https://archive.ics.uci.edu/dataset/94/spambase

[5] https://archive.ics.uci.edu/dataset/227/nomao

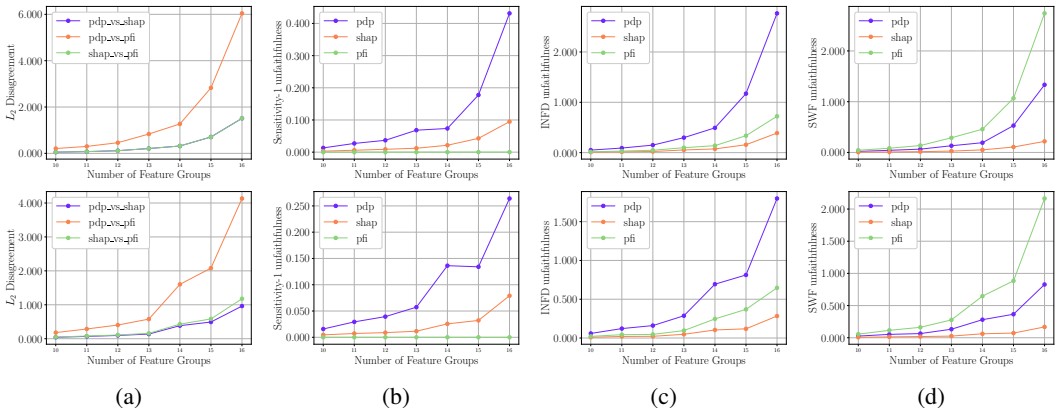

Figure 4: Running AGREED on the marketing tabular dataset: EBM model (top row) and GBT (bottom row). (a) $L_2$ disagreement between any explainer pairing is reduced as AGREED groups more features. (b-c-d) Three inconsistent (un)faithfulness metrics collectively converge to 0 as AGREED groups features.

highlights the role of between-group interactions in the disagreements between explainability methods. Moreover, Figures 4 (b-c-d) highlights the inherent inconsistencies between the ranking of (un)faithfulness metrics. For instance, when no features are grouped ($D = 16$), Sensitivity-1 claims that PFI is the most faithful explanation, while the other two metrics claim that SHAP is most faithful. Also note that INFD and SWF disagree on the ranking between PDP and PFI. By reducing the number of groups (*i.e.* grouping more features together), all (un)faithfulness metrics collectively converge to zero for either PDP, SHAP, or PFI.

Grouping features reduces the inconsistencies between explainability methods and (un)faithfulness metrics. Nevertheless, the resulting explanation must be interpreted with care since the joint-attribution of a group is a *multivariate* function of all features involved. Appendix D.2.2 presents practical examples of how to interpret the joint influence of groups containing at most three features.

## 5.2 IMAGES

Convolutional Neural Networks (CNN) remain a strong baseline across a wide range of image domain tasks. Since these models involve the composition on multiple non-linear spatial filters, it is unrealistic to find a single partition $\mathcal{P}$ w.r.t which $f$ is groupwise additive across all images from the dataset. Instead, by fixing an image of interest $\boldsymbol{x}$, a baseline $\boldsymbol{b}$, and setting $\mathcal{D} = \delta_{\boldsymbol{x}}$ and $\mathcal{B} = \delta_{\boldsymbol{b}}$, AGREED will search for a partition w.r.t which the model is group-wise additive in the region $\prod_{j=1}^{d}[b_j, x_j] \subset \mathcal{X}$. While this could leads to explanations with reduced disagreements and increased faithfulness, this also implies that AGREED must be rerun for each individual image $\boldsymbol{x}$ leading to a partition that is only valid for this image.

Moreover, since pixels have an inherent sense of proximity, we further restrict the partition to describe $D$ *path-connected* patches: any two pixels within a patch must be connected via a path spanning said patch. To produce such patches, group fusion is only performed in AGREED if two patches share a boundary. Image experiments ran on a NVIDIA GeForce RTX 3090 GPU with 25GiB of RAM.

### 5.2.1 KNOWN GROUND-TRUTHS ON SYNTHETIC DATA

We first experimented on a synthetic image dataset for which ground-truth partitions are known. The dataset consisted of $W \times W$ black and white images with a random rectangle drawn into them. Each image is labeled as $y = 1$ if the rectangle is tall and $y = 0$ if it is wide. This classification problem can be solved exactly with a CNN that composes a convolution layer (with $2 \times 2$ filters that detect left, right, up, and down edges), a ReLU non-linearity, and a linear layer (see Figure 5 (a)). Although this model is not groupwise additive over its whole domain, it is groupwise additive in

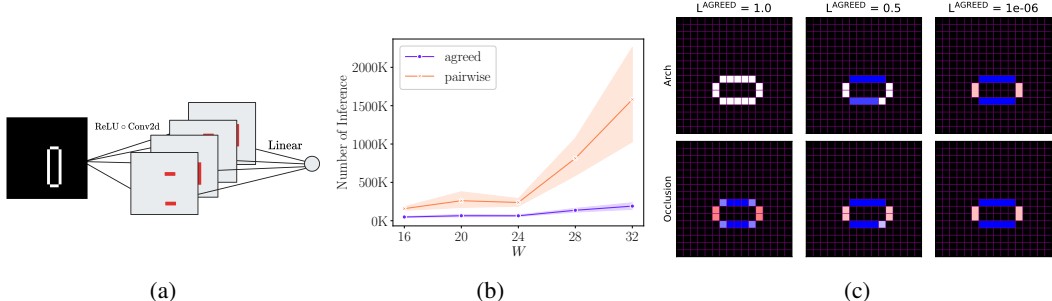

(a)                              (b)                              (c)

Figure 5: Synthetic image example. (a) A toy CNN can detect vertical and horizontal edges to perfectly classify rectangles while being locally group-wise additive. (b) Comparing the scalability of AGREED and PAIRWISE as the image size $W$ increases. (c) AGREED finds four patches to increase agreement between Arch and Occlusion.

the region $\prod_{j=1}^{d}[0, x_j]$ and the optimal partition $\mathcal{P}^{\star}$ considers each of the four rectangle edges as a separate group. Fixing the baseline $\boldsymbol{b} = \boldsymbol{0}$, we ran the AGREED and PAIRWISE methods (IGREEDY and Recurse were not developed for images), and confirmed that they both converge to the optimal partition for any image $\boldsymbol{x}$.

However, the PAIRWISE approach is much more computationally expensive than AGREED, as evidenced by Figure 5 (b). Indeed, PAIRWISE scales poorly w.r.t $W$ compared to AGREED. This is because the number of pairwise interactions to consider is $\mathcal{O}(W^4)$. AGREED avoids this complexity by making the assumption that pixel interactions in a CNN model are local. This assumption leads to efficient convergence on this synthetic data and model, see Figure 5 (c) for a qualitative example comparing Arch and Occlusion. Appendix D.3 illustrates how other explainability methods also reach perfect agreement.

### 5.2.2 SALIENCY MAPS ON MINIIMAGENET

We studied the VGG16, ResNet18, and ConvNext models pre-trained on ImageNet and explained their predictions on the MiniImageNet subset containing 100 classes and 600 images per class (Ravi & Larochelle, 2017). Given the lack of ground-truth partitions, we compared pixel partitioning algorithms in terms of their tradeoffs between disagreement/(un)faithfulness and patch size. The PAIRWISE grouping algorithm was not investigated because, on a realistic CNN, the resulting patches are no longer guaranteed to be path-connected. Instead, we compared AGREED with two algorithms previously used on ImageNet: the QUICKSHIFT image segmentation algorithm used by LIME and ArchAttribute, and $W \times W$ SQUARE patches implicit to Occlusion and RISE. To accelerate AGREED, we started with a partition of small $14 \times 14$ patches. A null baseline $\boldsymbol{b} = \boldsymbol{0}$ was used to mask out pixels. AGREED took 1-7 seconds per image to generate a partition for VGG16/ResNet18 and 4-30 seconds for ConvNext.

To compute a single explanation disagreement scores that can be compared across pixel grouping methods, we averaged disagreements over 100 randomly chosen test images $\boldsymbol{x}$ and over all pairings of the Arch/Occ/IG/LIME/SHAP explainers. RISE was excluded because it is equivalent to LIME. The (un)faithfulness metrics were also estimated using the same 100 test set images. Results of the experiments on the three models are presented in Figure 6. We see that AGREED offers the most competitive tradeoffs: lower disagreements/(un)faithfulness for smaller patches on average. In the case of ConvNext, AGREED is the only method that is able to consistently reduce disagreements/(un)faithfulness. For the other two methods, these quantities can either stagnate or even increase as pixel patches grow in size. We suspect this occurs because stronger feature interactions are involved in ConvNext compared to the other two models. Grouping the pixels involved in said interactions appear to be necessary to obtain saliency maps that agree with each other and that are more faithfull.

Appendix D.4 presents saliency maps yielded by AGREED, which tend to generate one large important patch that covers the object being classified, although there are exceptions. This suggests that, when explaining the function $f$ that maps pixels to class logit, all pixels that cover the object interact

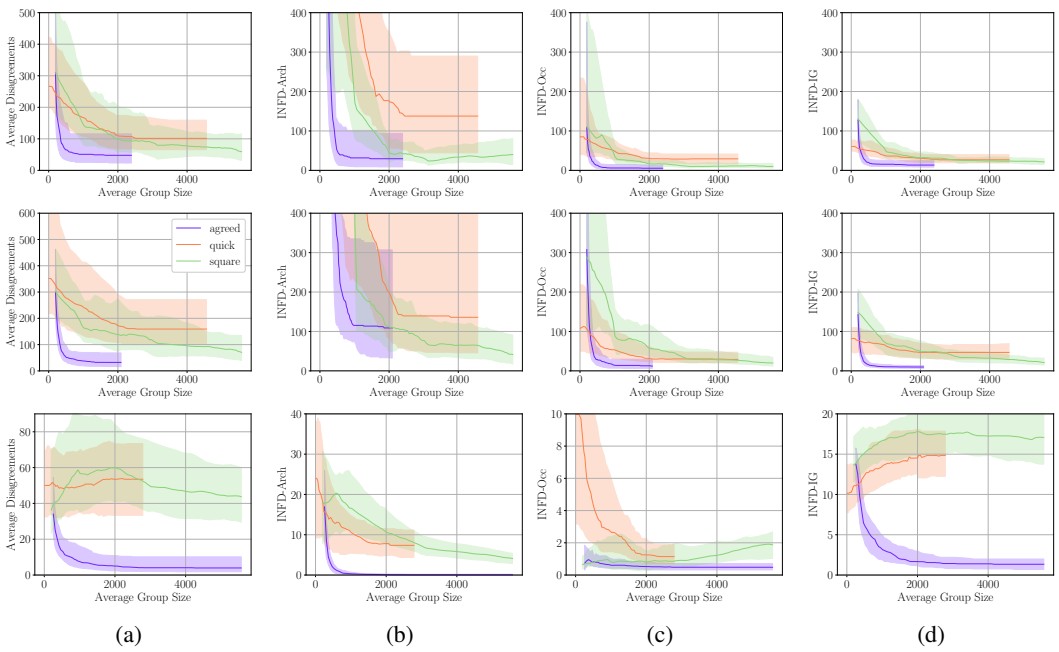

Figure 6: Running AGREED, QUICKSHIFT, and SQUARE on CNNs pre-trained on ImageNet: VGG16 (top row), ResNet18 (middle row), and ConvNext tiny (bottom row). (a) The average Arch/Occ/LIME/SHAP/IG $L_2$ disagreement versus the average size of pixel patches. (b-c-d) The INFD (un)faithfulness metric of Arch, Occ, and IG for various sizes of pixel groups. AGREED offers the most competitive trade-offs: lower disagreements/(un)faithfulness for smaller patches.

strongly with its neighbors. Since AGREED is constrained to output a partition of pixels (no overlap), pixels covering different semantic elements (*e.g.* different object parts) must be fused to ensure that no interactions are remaining. In future work, we envision extending AGREED to return overlapping pathces by applying the orignal algorithm on each the functions $f$ that map input pixels to concept activations in hidden layers (Fel et al., 2023). Explaining the input-concept mapping (instead of the input-logit one) might also allow for more semantically meaningful explanations when the related concepts are interpretable by humans.

## 6 CONCLUSION

We unified feature groups explanation methods through Functional Decomposition. We identified the culprit that prohibits agreement among the methods and (un)faithfulness metrics: between-group features interactions. The strength of these interactions was minimized using an novel algorithm, AGREED, that iteratively fuses feature groups whenever they interact. On two data modalities, AGREED was demonstrated to efficiently reduce inconsistencies between explanation techniques and (un)faithfulness metrics. AGREED is broadly applicable to tabular and image data although both structures are treated differently in the algorithm. Future efforts could extend it to other modalities where groups are natural, for example time-series and text.

For tabular data, the remaining challenge is to provide an automatic methodology for visualizing the joint attribution of feature groups involving more than three features. We envision combining AGREED with regional based explanations (Laberge et al., 2024) to interpret these high-dimensional functions regionally. For images classification, we managed to increase agreement betwee the ArchAttribute/Occlusion/IG/RISE/LIME/SHAP saliency maps. In most cases, the pixel grouping algorithm returns a single patch of upmost importance that covers the whole object. This clarifies "where" the network is looking but not "what" semantic elements the network is seeing. Combining AGREED with concept-based techniques (Fel et al., 2023) could potentially make explanations more semantically meaningful and might also lead to smaller pixel patches since overlap is allowed between patches related to different concepts.

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

# A    EXTENDED BACKGROUND

## A.1    NOTATION TABLE

| Notation | Definition |
|---|---|
| **Sets and Partitions** | |
| $d$ | Number of input features. |
| $[d] := \{1, 2, \ldots, d\}$ | Set of all features indices. |
| $j \in [d]$ | $j$ is an input feature index. |
| $u \subseteq [d]$ | $u$ is a subset of input features indices. |
| $D$ | Number of features groups. |
| $[D] := \{1, 2, \ldots, D\}$ | Set of all groups indices. |
| $i \in [D]$ | $i$ is a group index. |
| $U \subseteq [D]$ | $U$ is a subset of groups indices. |
| $\mathcal{P} : [d] \to [D]$ | Partition of $[d]$ into $D$ disjoint groups. |
| $\mathcal{P}(u) := \{\mathcal{P}(j) : j \in u\}$ | Partition mapping of feature subset $u$. |
| $\mathcal{P}^{-1}(U) := \{j \in [d] : \mathcal{P}(j) \in U\}$ | Partition inverse map of the group subset $U$. |
| **Explanations and Cooperative Game** | |
| $\mathcal{X} \subseteq \mathbb{R}^d$ | Input domain. |
| $\boldsymbol{x} \in \mathcal{X}$ | Input to explain. |
| $\mathcal{D}$ | Probability distribution for $\boldsymbol{x} \sim \mathcal{D}$. |
| $\boldsymbol{b} \in \mathcal{X}$ | Baseline used as a reference. |
| $\mathcal{B}$ | Probability distribution for $\boldsymbol{b} \sim \mathcal{B}$. Sometimes equal to $\mathcal{D}$. |
| $f : \mathcal{X} \to \mathbb{R}$ | Model to explain. |
| $\boldsymbol{r}_u : \mathcal{X} \times \mathcal{X} \to \mathcal{X}$ | Replace-function. |
| $f(\boldsymbol{x}) = \sum_{u \subseteq [d]} f_u(\boldsymbol{x})$ | Functional Decomposition. |
| $f_{u,\mathcal{B}}(\boldsymbol{x})$ | Marginal Decomposition. |
| $\phi(h, \boldsymbol{x}, \mathcal{B}, \mathcal{P}) \in \mathbb{R}^D$ | Group Importance toward the gap $f(\boldsymbol{x}) - \mathbb{E}_{\boldsymbol{b} \sim \mathcal{B}}[f(\boldsymbol{b})]$. |
| $\nu_{f,\boldsymbol{x},\mathcal{B},\mathcal{P}}$ | Coalitional game. |
| $\Delta_{f,\boldsymbol{x},\mathcal{B},\mathcal{P}}$ | Harsanyi Dividend. |
| $\mathcal{L}_f(\mathcal{D}, \mathcal{B}, \mathcal{P}) \in \mathbb{R}_+$ | Partition loss that depends on $\mathcal{D}$, $\mathcal{B}$, and $\mathcal{P}$. |

Table 1: All notation used throughout the paper.

## A.2    ANOVA DECOMPOSITION

When the baseline distribution represents independent features (*i.e.* $\mathcal{B} = \mathcal{B}_{\text{ind}} := \prod_{i=1}^{d} \mathcal{B}_i$), the Marginal Decomposition (cf. Def 2.1) falls back to the ANOVA decomposition (Hooker, 2004). This functional decomposition enjoys additional theoretical properties that do not necessarily hold for the Marginal Decomposition. Notably, the components $f_{u,\mathcal{B}_{\text{ind}}}$ are zero mean and uncorrelated

$$u \neq \emptyset \Rightarrow \mathbb{E}_{\boldsymbol{x} \sim \mathcal{B}_{\text{ind}}}[f_{u,\mathcal{B}_{\text{ind}}}(\boldsymbol{x})] = 0. \tag{15}$$

$$u \neq v \Rightarrow \mathbb{E}_{\boldsymbol{x} \sim \mathcal{B}_{\text{ind}}}[f_{u,\mathcal{B}_{\text{ind}}}(\boldsymbol{x})\, f_{v,\mathcal{B}_{\text{ind}}}(\boldsymbol{x})] = 0. \tag{16}$$

Letting $\sigma_u^2 := \mathbb{E}_{\boldsymbol{x} \sim \mathcal{B}_{\text{ind}}}[f_{u,\mathcal{B}_{\text{ind}}}(\boldsymbol{x})^2]$, the total variance of the model can be decomposed

$$\mathbb{E}_{\boldsymbol{x} \sim \mathcal{B}_{\text{ind}}}\big[\,(f(\boldsymbol{x}) - f_{\emptyset,\mathcal{B}_{\text{ind}}})^2\big] = \sum_{\substack{u \subseteq [d] \\ |u| \geq 1}} \sigma_u^2. \tag{17}$$

This property is where the terminology ANalysis Of VAriance comes from. Note that this elegant decomposition of model variance is not guaranteed to holds if features are non-independent. In that case, Equation 17 can potentially involve negative/positive correlation terms between $f_u$ and $f_v$. Since feature independence is unlikely to hold on realistic Machine Learning datasets, the ANOVA decomposition will be of solely theoretical interest : it is used to derive Property 3 of Definition 3.1.

### A.3 UNFAITHFULNESS METRICS

#### A.3.1 NO GROUP OVERLAP

Many unfaithfulness metrics proposed in the literature can be framed as

$$\underline{F}(\boldsymbol{\phi}, f, w) = \underset{\boldsymbol{x} \sim \mathcal{D}}{\mathbb{E}} \left[ \sum_{U \subseteq [D]} w(U) \left( \sum_{i \in U} \phi_i(f, \boldsymbol{x}, \mathcal{B}, \mathcal{P}) - [\nu_{f,\boldsymbol{x},\mathcal{B},\mathcal{P}}(U) - \nu_{f,\boldsymbol{x},\mathcal{B},\mathcal{P}}(\emptyset)] \right)^2 \right] \quad (18)$$

or

$$\overline{F}(\boldsymbol{\phi}, f, w) = \underset{\boldsymbol{x} \sim \mathcal{D}}{\mathbb{E}} \left[ \sum_{U \subseteq [D]} w(U) \left( \sum_{i \in U} \phi_i(f, \boldsymbol{x}, \mathcal{B}, \mathcal{P}) - [\nu_{f,\boldsymbol{x},\mathcal{B},\mathcal{P}}([D]) - \nu_{f,\boldsymbol{x},\mathcal{B},\mathcal{P}}([D] \setminus U)] \right)^2 \right] \quad (19)$$

using a distinct weight $w(U)$ for every coalition $U \subseteq [D]$. The Sensitivity-n metric proposed by Ancona et al. (2017) and the $\mu$-Fidelity of Bhatt et al. (2020) report the $\overline{F}$ metric using the weight $w(U) = \binom{D}{|U|}^{-1}$ for $|U| = n$ and $w(U) = 0$ otherwise. The INFD metric of Yeh et al. (2019) also employs the $\overline{F}$ metric but it weights the various coalition uniformly: $w(U) = 1/2^D$.

The Shapley-weighted Fidelity (Muschalik et al., 2025) reports the $\underline{F}$ score using the weights $w(\emptyset) = w(D) = \infty$ and $w(U) \propto \binom{D-2}{|U|-1}^{-1}$. The infinite weights can be seen as a hard constraint that the attribution should respect the *efficiency axiom*: they should sum up to the model prediction (Tsai et al., 2023). Nevertheless, for the purpose of comparing different explanations quality, we will be ignoring the edge cases $U = \emptyset, D$ when computing this faithfulness score.

#### A.3.2 WITH GROUP OVERLAP

There is on-going research on how to define Self-Attributing Neural Networks (SANN) that have a *build-in* notion of feature group importance. This is typically done by restricting the model architecture to $f(\boldsymbol{x}) = \sum_{i=1}^{D} f_i(\boldsymbol{x}_{G_i})$, where $G_i \subseteq [d]$ is the $i$th group (You et al., 2025). The proposed build-in patch importance is then $\phi_i^{\mathrm{SANN}}(f, \boldsymbol{x}) := f_i(\boldsymbol{x}_{G_i})$. Since groups can overlap (*i.e.* $G_i \cap G_k \neq \emptyset$), the unfaithfulness metrics discussed in the previous section are not applicable to these explanations.

The literature on SANN uses alternative measures of unfaithfulness. Fixing $\mathcal{B} = \delta_{\boldsymbol{0}}$ to a null image, letting $S \subseteq [d]$ be a subset of features of interest, and letting $(G_1, G_2, \ldots, G_D)$ be $D$ feature groups $G_i \subseteq [d]$ that possibly overlap, the Insertion and Deletion errors are defined as (You et al., 2025)

$$\mathrm{InsErr}(f, \phi, S) = \left| f(\boldsymbol{r}_S(\boldsymbol{0}, \boldsymbol{x})) - f(\boldsymbol{0}) - \sum_{i \leq D : G_i \subseteq S} \phi_i(f, \boldsymbol{x}) \right| \quad (20)$$

$$\mathrm{DelErr}(f, \phi, S) = \left| f(\boldsymbol{x}) - f(\boldsymbol{r}_S(\boldsymbol{x}, \boldsymbol{0})) - \sum_{i \leq D : G_i \cap S \neq \emptyset} \phi_i(f, \boldsymbol{x}) \right|. \quad (21)$$

These two definitions are generalizations of Equations 19 & 18 that allow for overlap between groups. Indeed, when the groups $G_i$ happen to define a partition of $[d]$, then these new definitions fall back to the previous ones.

This generalization does not come for free though. Theorem 3.2 highlights that being group-wise additive w.r.t to a partition of $[d]$ is sufficient to reach optimality w.r.t Equations 19 & 18. Attaining an unfaithfulness of 0 according to Equations 20 & 21 requires different assumptions about the model. You et al. (2025) proved that the InsErr and DelErr of SANN explanations are zero whenever the model takes the form

$$f(\boldsymbol{x}) = \sum_{i=1}^{D} p_i(\boldsymbol{x}_{G_i}), \quad \text{with} \quad p_i(\boldsymbol{x}_{G_i}) = \sum_{k=1}^{K_i} a_{ik} \prod_{j \in G_i} x_j^{b_{ijk}}. \quad (22)$$

where $a_{ij} \in \mathbb{R}$ are coefficients of the polynomial and $b_{ijk} \in \{1, 2, 3, \ldots\}$ is the exponent for feature $j$ in group $i$ in the polynomial term indexed by $k$. It is crucial that all $x_j$ within a group $j \in G_i$

have a non-null exponent. This structure is respected by the SANN $f(\boldsymbol{x}) = x_1 x_2 - x_2 x_3$ with built-in groups $G_1 = \{1, 2\}$ and $G_2 = \{2, 3\}$. The corresponding InsErr and DelErr scores using $S = \{1, 2\}$ and $\boldsymbol{x} = (1, 1, 1)$ are

$$\text{InsErr}(f, \phi^{\text{SANN}}, S) = |f((1,1,0)) - f((0,0,0)) - \phi_1^{\text{SANN}}(f, \boldsymbol{x})| = |1 - 0 - 1| = 0. \tag{23}$$

$$\text{DelErr}(f, \phi^{\text{SANN}}, S) = |f((1,1,1)) - f((0,0,1)) - \phi_1^{\text{SANN}}(f, \boldsymbol{x}) - \phi_2^{\text{SANN}}(f, \boldsymbol{x})| = |2 - 0 - 1 - 1| = 0. \tag{24}$$

This confirms that SANN explanations are the most faithful explanations of $f$ according to both metrics. However, if the model does not respect Equation 22, *e.g.* $f(x) = \sin(x_1 + x_2) + \sin(x_2 + x_3)$, using the same groups $G_1 = \{1, 2\}$ and $G_2 = \{2, 3\}$ no longer leads to a null InsErr

$$\text{InsErr}(f, \phi^{\text{SANN}}, S) = |f((1,1,0)) - f((0,0,0)) - \phi_1^{\text{SANN}}(f, \boldsymbol{x})| = |\sin(2) + \sin(1) - 0 - \sin(2)| \neq 0. \tag{25}$$

In short, whether or not one should consider overlapping vs non-overlapping feature groups may depend on the model. Non-overlapping groups may be preferable when $f$ is assumed to be (approximately) group-wise additive. Overlapping groups may be more pertinent if the model can be (approximately) expressed in the polynomial form outlined in Equation 22.

## B  PROOFS

### B.1  HARSANYI DIVIDENDS

Before unifying the various post-hoc explainers proposed in the literature, we must introduce an additional definition.

**Definition B.1** (Harsanyi Dividends (Harsanyi, 1963)). *Given a coalitional game $\nu$ involving $D$ players, its Harsanyi Dividend $\Delta$ is defined recursively*

$$\Delta(U) := \nu(U) - \sum_{V \subset U} \Delta(V), \tag{26}$$

*for any coalition $U \subseteq [D]$.*

The base cases are $\Delta(\emptyset) = \nu(\emptyset)$ and $\Delta(\{i\}) = \nu(\{i\}) - \nu(\emptyset)$.

Harsanyi dividends $\Delta(U)$ can be interpreted as the *excess gain* of a coalition $\nu(U) - \nu(\emptyset)$ that cannot be explained by any cooperation between strict non-empty subsets $V \subset U$. We can reorganize Equation 26 to express any coalitional game in terms of its dividends

$$\nu(U) = \sum_{V \subseteq U} \Delta(V) \quad \text{for any } U \subseteq [D]. \tag{27}$$

Similarily, we can reexpress the definition of the Marginal Decomposition (cf. Definition 2.1)

$$\mathbb{E}_{\boldsymbol{b} \sim \mathcal{B}}[f(\boldsymbol{r}_u(\boldsymbol{b}, \boldsymbol{x}))] = \sum_{v \subseteq u} f_{v,\mathcal{B}}(\boldsymbol{x}) \quad \text{for any } u \subseteq [d]. \tag{28}$$

The ressemblance between Equations 27 and 28 is striking and suggests a deeper connection between the Marginal Decomposition and the Harsanyi Dividend. Our goal with this subsection is to highlight the link between these two related (but different) concepts.

**Lemma B.1.** *Given a partition $\mathcal{P} : [d] \to [D]$, the following holds*

$$u \subseteq \mathcal{P}^{-1}(U) \iff \mathcal{P}(u) \subseteq U, \tag{29}$$

*where $u \subseteq [d]$ and $U \subseteq [D]$ are subsets of features and feature groups respectivelly.*

*Proof.* We start from this simple consequence of the definitions of $\mathcal{P}$

$$j \in \mathcal{P}^{-1}(U) \iff \mathcal{P}(j) \in U. \tag{30}$$

The goal of the lemma is to translate this equivalence to feature subsets $u$ and not just a single feature $j$. Letting $u \subseteq [d]$, we have

$$u \subseteq \mathcal{P}^{-1}(U) \iff \forall i \in u : i \in \mathcal{P}^{-1}(U) \iff \forall i \in u : \mathcal{P}(i) \in U \iff \mathcal{P}(u) \subseteq U. \tag{31}$$

$\square$

**Lemma B.2.** *The Harsanyi Dividend $\Delta_{f,\boldsymbol{x},\mathcal{B},\mathcal{P}}$ for the coalitional game of Definition 2.3 can be expressed in terms of the Marginal Decomposition*

$$\Delta_{f,\boldsymbol{x},\mathcal{B},\mathcal{P}}(U) = \sum_{u \subseteq [d]:\mathcal{P}(u)=U} f_{u,\mathcal{B}}(\boldsymbol{x}), \tag{32}$$

*under the convention that $\mathcal{P}(\emptyset) = \emptyset$.*

*Proof.* The proof proceeds by induction. The base case covers all dividends $\Delta_{f,\boldsymbol{x},\mathcal{B},\mathcal{P}}(U)$ such that $|U| \leq 1$. Indeed, we have:

$$\Delta_{f,\boldsymbol{x},\mathcal{B},\mathcal{P}}(\emptyset) := \nu_{f,\boldsymbol{x},\mathcal{B},\mathcal{P}}(\emptyset) := \mathbb{E}_{\boldsymbol{b} \sim \mathcal{B}}[f(\boldsymbol{b})] = f_{\emptyset,\mathcal{B}}(\boldsymbol{x}). \tag{33}$$

and

$$
\begin{aligned}
\Delta_{f,\boldsymbol{x},\mathcal{B},\mathcal{P}}(\{i\}) :=& \nu_{f,\boldsymbol{x},\mathcal{B},\mathcal{P}}(\{i\}) - \nu_{f,\boldsymbol{x},\mathcal{B},\mathcal{P}}(\emptyset) \\
:=& \operatorname*{\mathbb{E}}_{\boldsymbol{b}\sim\mathcal{B}}[f(\boldsymbol{r}_{\mathcal{P}^{-1}(\{i\})}(\boldsymbol{b},\boldsymbol{x}))] - \operatorname*{\mathbb{E}}_{\boldsymbol{b}\sim\mathcal{B}}[f(\boldsymbol{b})] && \text{(cf. Definition 2.3)} \\
=& \sum_{u\subseteq\mathcal{P}^{-1}(\{i\})} f_{u,\mathcal{B}}(\boldsymbol{x}) - f_{\emptyset,\mathcal{B}}(\boldsymbol{x}) && \text{(cf. Equation 28)} \\
=& \sum_{u\subseteq[d]:\mathcal{P}(u)\subseteq\{i\}} f_{u,\mathcal{B}}(\boldsymbol{x}) - f_{\emptyset,\mathcal{B}}(\boldsymbol{x}) && \text{(cf. Lemma B.1)} \\
=& \sum_{u\subseteq[d]:\mathcal{P}(u)=\{i\}} f_{u,\mathcal{B}}(\boldsymbol{x}). &&
\end{aligned}
$$

Now fixing $U \subseteq [D]$ and assuming the premise holds for all $V \subset U$, we have

$$
\begin{aligned}
\Delta_{f,\boldsymbol{x},\mathcal{B},\mathcal{P}}(U) =& \nu_{f,\boldsymbol{x},\mathcal{B},\mathcal{P}}(U) - \sum_{V\subset U}\Delta_{f,\boldsymbol{x},\mathcal{B},\mathcal{P}}(V) && \text{(cf. Definition B.1)} \\
=& \operatorname*{\mathbb{E}}_{\boldsymbol{b}\sim\mathcal{B}}[f(\boldsymbol{r}_{\mathcal{P}^{-1}(U)}(\boldsymbol{b},\boldsymbol{x}))] - \sum_{V\subset U}\Delta_{f,\boldsymbol{x},\mathcal{B},\mathcal{P}}(V) && \text{(cf. Definition 2.3)} \\
=& \sum_{u\subseteq\mathcal{P}^{-1}(U)} f_{u,\mathcal{B}}(\boldsymbol{x}) - \sum_{V\subset U}\Delta_{f,\boldsymbol{x},\mathcal{B},\mathcal{P}}(V) && \text{(cf. Equation 28)} \\
=& \sum_{u\subseteq[d]:\mathcal{P}(u)\subseteq U} f_{u,\mathcal{B}}(\boldsymbol{x}) - \sum_{V\subset U}\Delta_{f,\boldsymbol{x},\mathcal{B},\mathcal{P}}(V) && \text{(cf. Lemma B.1)} \\
=& \sum_{u\subseteq[d]:\mathcal{P}(u)\subseteq U} f_{u,\mathcal{B}}(\boldsymbol{x}) - \sum_{V\subset U}\sum_{u\subseteq[d]:\mathcal{P}(u)=V} f_{u,\mathcal{B}}(\boldsymbol{x}) && \\
&&& \text{(By the recursion Assumption)} \\
=& \sum_{u\subseteq[d]:\mathcal{P}(u)\subseteq U} f_{u,\mathcal{B}}(\boldsymbol{x}) - \sum_{u\subseteq[d]:\mathcal{P}(u)\subset U} f_{u,\mathcal{B}}(\boldsymbol{x}) && \\
=& \sum_{u\subseteq[d]:\mathcal{P}(u)=U} f_{u,\mathcal{B}}(\boldsymbol{x}) &&
\end{aligned}
$$

concluding the proof.

$\square$

## B.2 UNIFICATION

**Theorem B.1** (**Theorem 3.1**). *The Arch/Occ/LIME/SHAP/RISE Attributions following can be expressed via the Marginal Decomposition*

$$\phi_i^\mu(\boldsymbol{x}, f, \mathcal{B}, \mathcal{P}) = \sum_{u \subseteq [d]:\{i\}=\mathcal{P}(u)} f_{u,\mathcal{B}}(\boldsymbol{x}). + \sum_{u \subseteq [d]:\{i\}\subsetneq\mathcal{P}(u)} h(|\mathcal{P}(u)|)f_{u,\mathcal{B}}(\boldsymbol{x}), \qquad (34)$$

*where $h$ is different for each $\mu$. Different explainability methods disagree on how to redistribute **between-group interactions** ($f_{u,\mathcal{B}}(\boldsymbol{x})$) with $\mathcal{P}(u) \geq 2$ among the groups involved.*

*Proof.* The proof of the theorem consists of expressing the attributions in terms of the Harsanyi Dividend $\Delta_{f,\boldsymbol{x},\mathcal{B},\mathcal{P}}$ (cf. Definition B.1), 2) Use Lemma B.2 to express the Dividend in terms of the Marginal Decomposition.

**ArchAttribute/Joint-PDP** can be expressed in terms of the Harsanyi Dividend

$$\phi_i^{\text{Arch}}(f, \boldsymbol{x}, \mathcal{B}, \mathcal{P}) = \Delta_{f,\boldsymbol{x},\mathcal{B},\mathcal{P}}(\{i\}). \qquad (35)$$

Following Lemma B.2 its holds that

$$\phi_i^{\text{Arch}}(f, \boldsymbol{x}, \mathcal{B}, \mathcal{P}) = \sum_{u \subseteq [d]:\mathcal{P}(u)=\{i\}} f_{u,\mathcal{B}}(\boldsymbol{x}). \qquad (36)$$

**Patch-Occlusion/joint-PFI** is also easily expressed in terms of Harsanyi Dividends

$$\begin{aligned}
\phi_i^{\text{Occ}}(f, \boldsymbol{x}, \mathcal{B}, \mathcal{P}) &= \nu_{f,\boldsymbol{x},\mathcal{B},\mathcal{P}}([D]) - \nu_{f,\boldsymbol{x},\mathcal{B},\mathcal{P}}([D] \setminus \{i\}) \\
&= \sum_{U \subseteq [D]} \Delta_{f,\boldsymbol{x},\mathcal{B},\mathcal{P}}(U) - \sum_{U \subseteq [D]\setminus\{i\}} \Delta_{f,\boldsymbol{x},\mathcal{B},\mathcal{P}}(U) \qquad \text{(cf. Equation 27)} \\
&= \sum_{U \subseteq [D]:i\in U} \Delta_{f,\boldsymbol{x},\mathcal{B},\mathcal{P}}(U) \\
&= \sum_{U \subseteq [D]:i\in U} \sum_{u \subseteq [d]:\mathcal{P}(u)=U} f_{u,\mathcal{B}}(\boldsymbol{x}) \qquad \text{(cf. Lemma B.2)} \\
&= \sum_{u \subseteq [d]:i\in\mathcal{P}(u)} f_{u,\mathcal{B}}(\boldsymbol{x}).
\end{aligned}$$

**Shapley Values** are known to redistribute the Harsanyi Dividends evenly between all players involved (Shapley, 1953)

$$\begin{aligned}
\phi_i^{\text{SHAP}}(f, x, \mathcal{B}, \mathcal{P}) &= \sum_{U \subseteq [D]:i\in U} \Delta_{f,\boldsymbol{x},\mathcal{B},\mathcal{P}}(U)/|U| \qquad \text{((Shapley, 1953))} \\
&= \sum_{U \subseteq [D]:i\in U} |U|^{-1} \sum_{u \subseteq [d]:\mathcal{P}(u)=U} f_{u,\mathcal{B}}(\boldsymbol{x}) \qquad \text{(cf. Lemma B.2)} \\
&= \sum_{U \subseteq [D]:i\in U} \sum_{u \subseteq [d]:\mathcal{P}(u)=U} f_{u,\mathcal{B}}(\boldsymbol{x})/|\mathcal{P}(u)| \qquad \text{(Since $|U| = |\mathcal{P}(u)|$)} \\
&= \sum_{u \subseteq [d]:i\in\mathcal{P}(u)} f_{u,\mathcal{B}}(\boldsymbol{x})/|\mathcal{P}(u)|.
\end{aligned}$$

**RISE** computes

$$\phi_i^{\text{RISE}}(f, \boldsymbol{x}, \mathcal{B}, \mathcal{P}) := \frac{1}{2^{D-1}} \sum_{U \subseteq [D]\setminus\{i\}} \nu_{f,\boldsymbol{x},\mathcal{B},\mathcal{P}}(U \cup \{i\}) - \nu_{f,\boldsymbol{x},\mathcal{B},\mathcal{P}}(U), \qquad (37)$$

which is actually the Banzhaf Index (Marichal et al., 2007). It is well-established that this attribution method assigns a score to each player by sharing Harsanyi Dividends using a power-of-two rule

$$\phi_i^{\text{RISE}}(f, \boldsymbol{x}, \mathcal{B}, \mathcal{P}) = \sum_{U \subseteq [D]: i \in U} \Delta_{f,\boldsymbol{x},\mathcal{B},\mathcal{P}}(U)/2^{|U|-1} \quad \text{(See Page 8 from (Marichal et al., 2007))}$$

$$= \sum_{U \subseteq [D]: i \in U} 1/2^{|U|-1} \sum_{u \subseteq [d]: \mathcal{P}(u)=U} f_{u,\mathcal{B}}(\boldsymbol{x}) \quad \text{(cf. Lemma B.2)}$$

$$= \sum_{U \subseteq [D]: i \in U} \sum_{u \subseteq [d]: \mathcal{P}(u)=U} f_{u,\mathcal{B}}(\boldsymbol{x})/2^{|\mathcal{P}(u)|-1} \quad \text{(Since } |U| = |\mathcal{P}(u)|)$$

$$= \sum_{u \subseteq [d]: i \in \mathcal{P}(u)} f_{u,\mathcal{B}}(\boldsymbol{x})/2^{|\mathcal{P}(u)|-1}.$$

**LIME**  advocates fitting a linear model on the function output evaluated on masked inputs

$$(\omega_0, \omega_1, \omega_2, \ldots, \omega_D) = \underset{\boldsymbol{\omega} \in \mathbb{R}^{D+1}}{\operatorname{argmin}} \frac{1}{2^D} \sum_{U \subseteq [D]} \left( \nu_{f,\boldsymbol{x},\mathcal{B},\mathcal{P}}(U) - \omega_0 - \sum_{i=1}^{D} \omega_i \mathbb{1}[i \in U] \right)^2. \quad (38)$$

and reporting the coefficients $(w_1, w_2, \ldots, w_D)$ as the local feature-groups attributions. This minimization problem is an alternative formulation of the Banzhaf Index (Tsai et al., 2023) so LIME is equivalent to RISE.

We have thus proven that Arch/Occ/LIME/RISE/SHAP can be expressed as

$$\phi_i(f, \boldsymbol{x}, \mathcal{B}, \mathcal{P}) = \sum_{u \subseteq [d]: \mathcal{P}(u)=\{i\}} f_{u,\mathcal{B}}(\boldsymbol{x}) + \sum_{u \subseteq [d]: \{i\} \subsetneq \mathcal{P}(u)} h(|\mathcal{P}(u)|) f_{u,\mathcal{B}}(\boldsymbol{x}), \quad (39)$$

where $h(|\mathcal{P}(u)|) = 0$ for ArchAttribute, $h(|\mathcal{P}(u)|) = 1$ for Occlusion, $h(|\mathcal{P}(u)|) = 1/|\mathcal{P}(u)|$ for SHAP, and $h(|\mathcal{P}(u)|) = 1/2^{|\mathcal{P}(u)|-1}$ for RISE/LIME.

$\square$

Before highlighting the critical role of between-group interactions in the (un)faithfulness metrics, we first recall the Minimality property of the Marginal Decomposition.

**Corollary B.2 (Corollary A.1** from (Laberge et al., 2024)). *Let $R \subseteq \mathbb{R}^d$ be a hyperrectangle and let $f : \mathbb{R}^d \to \mathbb{R}$ be a function that can be written $f(\boldsymbol{x}) = \sum_{u \subseteq [d]} g_u(\boldsymbol{x}) \, \forall \boldsymbol{x} \in R$, where $g_u$ only depends on $\boldsymbol{x}_u$. Also, assume that a subset $v \subset [d]$ exists such that*

$$u \supseteq v \Rightarrow \forall \boldsymbol{x} \in R \ \ g_u(\boldsymbol{x}) = 0.$$

*Then, for any probability distribution $\mathcal{B}$ such that $supp(\mathcal{B}) \subseteq R$, the Marginal Decomposition respects*

$$u \supseteq v \Rightarrow \forall \boldsymbol{x} \in R \ \ f_{u,\mathcal{B}}(\boldsymbol{x}) = 0.$$

Minimality implies that the Marginal Decomposition will not contain interactions that are not present in the model in the first place. This theorem induces an important corollary.

**Lemma B.3.** *Let $R$ be a hyperrectangle region and let $f$ be groupwise additive in $R$ w.r.t partition $\mathcal{P}$. Then for any distribution $\mathcal{B}$ such that $supp(\mathcal{B}) \subseteq R$, it holds that*

$$|\mathcal{P}(v)| \geq 2, \boldsymbol{x} \in R \Rightarrow f_{v,\mathcal{B}}(\boldsymbol{x}) = 0.$$

*Proof.* Let $f$ be groupwise additive in $R$ w.r.t $\mathcal{P}$. Thus there exists a rectangular region $R$ such that $supp(\mathcal{B}) \subseteq R$, and $f$ can be written as

$$f(\boldsymbol{x}) = \omega_0 + \sum_{i=1}^{D} g_{\mathcal{P}^{-1}(\{i\})}(\boldsymbol{x}) \ \ \forall \boldsymbol{x} \in R.$$

Thus, $f$ can be written in the form $\sum_{u \subseteq [d]} g_u$ where $|\mathcal{P}(u)| \geq 2 \Rightarrow g_u = 0$.

Now, letting $\boldsymbol{x} \in R$ be an input and $v \subseteq [d]$ be any feature subset such that $|\mathcal{P}(v)| \geq 2$. Any superset $u \supseteq v$ respects $|\mathcal{P}(u)| \geq |\mathcal{P}(v)| \geq 2$ and so $u \supseteq v \Rightarrow g_u(\boldsymbol{x}) = 0$. By minimality (cf. Theorem B.2), this implies that $f_{v,\mathcal{B}}(\boldsymbol{x}) = 0$ and so

$$|\mathcal{P}(v)| \geq 2, \boldsymbol{x} \in R \Rightarrow f_{v,\mathcal{B}}(\boldsymbol{x}) = 0.$$

$\square$

With this corollary now proven, we can demonstrate how group-wise additive models have faithful explanations according to the various (un)faithfulness metrics.

**Theorem B.3** (**Theorem 3.2**). *Let $R \subseteq \mathcal{X}$ be a hyperrectangle region such that $supp(\mathcal{D}) \subseteq R$ and $supp(\mathcal{B}) \subseteq R$. Let $\mathcal{P}$ be a feature partition. Whenever the model $f$ is groupwise additive in $R$ w.r.t $\mathcal{P}$, (un)faithfulness metrics that follow Equations 10 and 11 are all simultaneously minimized*

$$F(\boldsymbol{\phi}^\mu, f, w) = 0 \tag{40}$$

*for any weight function $w$ and attribution $\boldsymbol{\phi}^\mu$.*

*Proof.* Fix the set $U \subseteq [D]$. The metrics $\overline{F}$ and $\underline{F}$ are simply weighted aggregates of the difference between $\sum_{i \in U} \phi_i^\mu(f, \boldsymbol{x}, \mathcal{B}, \mathcal{P})$ or $\nu_{f,\boldsymbol{x},\mathcal{B},\mathcal{P}}(U) - \nu_{f,\boldsymbol{x},\mathcal{B},\mathcal{P}}(\emptyset)$ or $\nu_{f,\boldsymbol{x},\mathcal{B},\mathcal{P}}([D]) - \nu_{f,\boldsymbol{x},\mathcal{B},\mathcal{P}}([D] \setminus U)$. By Theorem 3.1 and Lemma B.3, the first term is equal to

$$\sum_{i \in U} \phi_i^\mu(f, \boldsymbol{x}, \mathcal{B}, \mathcal{P}) = \sum_{i \in U} \sum_{u \subseteq [d]: \{i\} = \mathcal{P}(u)} f_{u,\mathcal{B}}(\boldsymbol{x})$$

$$= \sum_{u \subseteq [d]: \mathcal{P}(u) \subseteq U, |\mathcal{P}(u)| = 1} f_{u,\mathcal{B}}(\boldsymbol{x}).$$

By Corollary B.2, the other two terms are also equal to this quantity

$$\nu_{f,\boldsymbol{x},\mathcal{B},\mathcal{P}}(U) - \nu_{f,\boldsymbol{x},\mathcal{B},\mathcal{P}}(\emptyset) = \sum_{V \subseteq U} \Delta_{f,\boldsymbol{x},\mathcal{B},\mathcal{P}}(V) - \Delta_{f,\boldsymbol{x},\mathcal{B},\mathcal{P}}(\emptyset) \quad \text{(cf. Eq 27)}$$

$$= \sum_{V \subseteq U: V \neq \emptyset} \Delta_{f,\boldsymbol{x},\mathcal{B},\mathcal{P}}(V)$$

$$= \sum_{V \subseteq U: V \neq \emptyset} \sum_{u \subseteq [d]: \mathcal{P}(u) = V} f_{u,\mathcal{B}}(\boldsymbol{x}) \quad \text{(cf. Lemma B.2)}$$

$$= \sum_{u \subseteq [d]: \mathcal{P}(u) \subseteq U, \mathcal{P}(u) \neq \emptyset} f_{u,\mathcal{B}}(\boldsymbol{x})$$

$$= \sum_{u \subseteq [d]: \mathcal{P}(u) \subseteq U, |\mathcal{P}(u)| = 1} f_{u,\mathcal{B}}(\boldsymbol{x}) \quad \text{(cf. Lemma B.3)}$$

$$\nu_{f,\boldsymbol{x},\mathcal{B},\mathcal{P}}([D]) - \nu_{f,\boldsymbol{x},\mathcal{B},\mathcal{P}}([D] \setminus U) = \sum_{V \subseteq [D]} \Delta_{f,\boldsymbol{x},\mathcal{B},\mathcal{P}}(V) - \sum_{V \subseteq [D] \setminus U} \Delta_{f,\boldsymbol{x},\mathcal{B},\mathcal{P}}(V) \quad \text{(cf. Eq 27)}$$

$$= \sum_{V \subseteq [D]: V \cap U \neq \emptyset} \Delta_{f,\boldsymbol{x},\mathcal{B},\mathcal{P}}(V)$$

$$= \sum_{V \subseteq [D]: V \cap U \neq \emptyset} \sum_{u \subseteq [d]: \mathcal{P}(u) = V} f_{u,\mathcal{B}}(\boldsymbol{x}) \quad \text{(cf. Lemma B.2)}$$

$$= \sum_{u \subseteq [d]: \mathcal{P}(u) \cap U \neq \emptyset} f_{u,\mathcal{B}}(\boldsymbol{x})$$

$$= \sum_{u \subseteq [d]: \mathcal{P}(u) \cap U \neq \emptyset, |\mathcal{P}(u)| = 1} f_{u,\mathcal{B}}(\boldsymbol{x}) \quad \text{(cf. Lemma B.2)}$$

$$= \sum_{u \subseteq [d]: \mathcal{P}(u) \subseteq U, |\mathcal{P}(u)| = 1} f_{u,\mathcal{B}}(\boldsymbol{x}).$$

$\square$

## B.3 PARTITION LOSS FUNCTION

In this section, we demonstrate the various properties of partition loss functions, which will be used to infer feature groups. We start by presenting a link between the cardinalities $|\mathcal{P}(u)|$ and $|\mathcal{P}'(u)|$ when $\mathcal{P}'$ is a superpartition of $\mathcal{P}$

**Lemma B.4.** *Let $\mathcal{P}'$ be a superpartition of $\mathcal{P}$, then for any subset $u \subseteq [d]$ is holds that*

$$|\mathcal{P}(u)| \geq |\mathcal{P}'(u)|. \tag{41}$$

*Proof.* Recall the definition of superpartition: $\mathcal{P}'$ is a superpartition of $\mathcal{P}$ if $\mathcal{P}(i) = \mathcal{P}(j) \Rightarrow \mathcal{P}'(i) = \mathcal{P}'(j)$. Conversly, $\mathcal{P}'(i) \neq \mathcal{P}'(j) \Rightarrow \mathcal{P}(i) \neq \mathcal{P}(j)$ must hold. This implies that $|\mathcal{P}(u)| \geq |\mathcal{P}'(u)|$ for any $u \subseteq [d]$. To prove it, assume the opposite holds: $|\mathcal{P}(u)| < |\mathcal{P}'(u)|$. This implies the existence of $|\mathcal{P}'(u)|$ points $i, j \in u$ such that $\mathcal{P}'(i) \neq \mathcal{P}'(j)$. However, by the definition of superpartition, $\mathcal{P}(i) \neq \mathcal{P}(j)$ must also hold for these $|\mathcal{P}'(u)|$ points, which contradicts the assumption that $\mathcal{P}(u)$ has smaller cardinality than $\mathcal{P}'(u)$. $\square$

**Lemma B.5.** *Define the objective*

$$\mathcal{L}_f(\mathcal{D}, \mathcal{B}, \mathcal{P}) := \mathop{\mathbb{E}}_{\boldsymbol{x} \sim \mathcal{D}} \left[ \sum_{\substack{u,v \subseteq [d] \\ |\mathcal{P}(u)| \geq 2, |\mathcal{P}(v)| \geq 2}} w(\mathcal{P}(u), \mathcal{P}(v)) f_{u,\mathcal{B}}(\boldsymbol{x}) f_{v,\mathcal{B}}(\boldsymbol{x}) \right] \tag{42}$$

*for some function $w$ such that $w(\mathcal{P}(u), \mathcal{P}(u)) \geq w(\mathcal{P}'(u), \mathcal{P}'(u)) \geq 0$ for any interaction $u$. Then $\mathcal{L}_f$ respects Definition 3.1.*

*Proof.* We prove the function $\mathcal{L}_f$ from Equation 42 respects the three properties of **Definition 3.1**.

**Property 1**   Let $f$ be group-wise additive in $R$ w.r.t $\mathcal{P}$. Lemma B.3 states that, for any probability distribution such that $\mathrm{supp}(\mathcal{B}) \subseteq R$, $\mathrm{supp}(\mathcal{D}) \subseteq R$, the following holds

$$|\mathcal{P}(v)| \geq 2, \boldsymbol{x} \in R \Rightarrow f_{v,\mathcal{B}}(\boldsymbol{x}) = 0.$$

Sampled inputs $\boldsymbol{x} \sim \mathcal{D}$ are guaranteed to land in $R$ (since $\mathrm{supp}(\mathcal{D}) \subseteq R$) and so

$$\mathcal{L}_f(\mathcal{D}, \mathcal{B}, \mathcal{P}) = \mathop{\mathbb{E}}_{\boldsymbol{x} \sim \mathcal{D}} \left[ \sum_{\substack{u,v \subseteq [d] \\ |\mathcal{P}(u)| \geq 2, |\mathcal{P}(v)| \geq 2}} w(\mathcal{P}(u), \mathcal{P}(v)) f_{u,\mathcal{B}}(\boldsymbol{x}) f_{v,\mathcal{B}}(\boldsymbol{x}) \right] = \mathop{\mathbb{E}}_{\boldsymbol{x} \sim \mathcal{D}}[0] = 0.$$

**Property 2**   Given a partition $\mathcal{P}$ of $[d]$ into $D$ groups, assume w.l.o.g that $f$ is additive w.r.t group $D$. Also, assume we wish to fuse group $D$ with group $D-1$, which will lead to a super partition $\mathcal{P}'$ such that

$$\mathcal{P}'(i) = \mathcal{P}(i) \quad \forall i \in \mathcal{P}^{-1}(\{1, 2, \ldots, D-1\}), \tag{43}$$

but $\mathcal{P}'(i) = D-1 \quad \forall i \in \mathcal{P}^{-1}(\{D\})$. The loss under partition $\mathcal{P}$ can be written

$$
\begin{aligned}
\mathcal{L}_f(\mathcal{D}, \mathcal{B}, \mathcal{P}) &= \mathop{\mathbb{E}}_{\boldsymbol{x} \sim \mathcal{D}} \left[ \sum_{\substack{u,v \subseteq [d] \\ |\mathcal{P}(u)| \geq 2, |\mathcal{P}(v)| \geq 2}} w(\mathcal{P}(u), \mathcal{P}(v)) f_{u,\mathcal{B}}(\boldsymbol{x}) f_{v,\mathcal{B}}(\boldsymbol{x}) \right] \\
&= \mathop{\mathbb{E}}_{\boldsymbol{x} \sim \mathcal{D}} \left[ \sum_{\substack{u,v \subseteq [d] \\ |\mathcal{P}(u)| \geq 2, |\mathcal{P}(v)| \geq 2 \\ D \in \mathcal{P}(u) \text{ or } D \in \mathcal{P}(v)}} w(\mathcal{P}(u), \mathcal{P}(v)) f_{u,\mathcal{B}}(\boldsymbol{x}) f_{v,\mathcal{B}}(\boldsymbol{x}) \right. \\
&\qquad\qquad \left. + \sum_{\substack{u,v \subseteq [d] \\ |\mathcal{P}(u)| \geq 2, |\mathcal{P}(v)| \geq 2 \\ D \notin \mathcal{P}(u) \text{ and } D \notin \mathcal{P}(v)}} w(\mathcal{P}(u), \mathcal{P}(v)) f_{u,\mathcal{B}}(\boldsymbol{x}) f_{v,\mathcal{B}}(\boldsymbol{x}) \right].
\end{aligned} \tag{44}
$$

We prove that $\mathcal{L}_f(\mathcal{D}, \mathcal{B}, \mathcal{P}) = \mathcal{L}_f(\mathcal{D}, \mathcal{B}, \mathcal{P}')$ by rewriting both summation terms of Equation 44. For the first term, we exploit the fact that $f$ is additive in $R$ w.r.t group $D$, implying the existence of a rectangular region $R$ such that

$$f(\boldsymbol{x}) = g_{\mathcal{P}^{-1}(\{D\})}(\boldsymbol{x}) + g_{\mathcal{P}^{-1}(\{1,2\dots,D-1\})}(\boldsymbol{x}) \quad \forall \boldsymbol{x} \in R.$$

Hence, $f$ can be written in the form $\sum_{u \subseteq [d]} g_u(\boldsymbol{x})$ where $[|\mathcal{P}(u)| \geq 2$ and $D \in \mathcal{P}(u)$ and $\boldsymbol{x} \in R] \Rightarrow g_u(\boldsymbol{x}) = 0$. Letting $v \subseteq [d]$ be some feature subset such that $|\mathcal{P}(v)| \geq 2$ and $D \in \mathcal{P}(v)$. Any superset $u \supseteq v$ respects $|\mathcal{P}(u)| \geq |\mathcal{P}(v)| \geq 2$ and $D \in \mathcal{P}(u)$, thus $u \supseteq v$ and $\boldsymbol{x} \in R \Rightarrow g_u(\boldsymbol{x}) = 0$. By minimality (cf. Corrolary B.2), the component $f_{v,\mathcal{B}}(\boldsymbol{x}) = 0$ is null

$$|\mathcal{P}(v)| \geq 2, D \in \mathcal{P}(v), \boldsymbol{x} \in R \Rightarrow f_{v,\mathcal{B}}(\boldsymbol{x}) = 0. \tag{45}$$

Accordingly, the first summation term of Equation 44 is null

$$\mathbb{E}_{\boldsymbol{x} \sim \mathcal{D}} \left[ \sum_{\substack{u,v \subseteq [d] \\ |\mathcal{P}(u)| \geq 2, |\mathcal{P}(v)| \geq 2 \\ D \in \mathcal{P}(u) \text{ or } D \in \mathcal{P}(v)}} w(\mathcal{P}(u), \mathcal{P}(v)) f_{u,\mathcal{B}}(\boldsymbol{x}) f_{v,\mathcal{B}}(\boldsymbol{x}) \right] = \mathbb{E}_{\boldsymbol{x} \sim \mathcal{D}}[0] = 0. \tag{46}$$

By Lemma B.4, we also have that $|\mathcal{P}'(u)| \geq 2$ implies $|\mathcal{P}(u)| \geq 2$ and so

$$\mathbb{E}_{\boldsymbol{x} \sim \mathcal{D}} \left[ \sum_{\substack{u,v \subseteq [d] \\ |\mathcal{P}'(u)| \geq 2, |\mathcal{P}'(v)| \geq 2 \\ D \in \mathcal{P}(u) \text{ or } D \in \mathcal{P}(v)}} w(\mathcal{P}'(u), \mathcal{P}'(v)) f_{u,\mathcal{B}}(\boldsymbol{x}) f_{v,\mathcal{B}}(\boldsymbol{x}) \right] = \mathbb{E}_{\boldsymbol{x} \sim \mathcal{D}}[0] = 0. \tag{47}$$

As a result, the left-most terms of Equations 46 and 47 are equal.

Now tackling the second term of Equation 44. By Equation 43, for any $u \subseteq [d]$ such that $D \notin \mathcal{P}(u)$ it holds that $\mathcal{P}(u) = \mathcal{P}'(u)$. So for any $\boldsymbol{x}$ we have

$$\sum_{\substack{u,v \subseteq [d] \\ |\mathcal{P}(u)| \geq 2, |\mathcal{P}(v)| \geq 2 \\ D \notin \mathcal{P}(u) \text{ and } D \notin \mathcal{P}(v)}} w(\mathcal{P}(u), \mathcal{P}(v)) f_{u,\mathcal{B}}(\boldsymbol{x}) f_{v,\mathcal{B}}(\boldsymbol{x}) = \sum_{\substack{u,v \subseteq [d] \\ |\mathcal{P}'(u)| \geq 2, |\mathcal{P}'(v)| \geq 2 \\ D \notin \mathcal{P}(u) \text{ and } D \notin \mathcal{P}(v)}} w(\mathcal{P}'(u), \mathcal{P}'(v)) f_{u,\mathcal{B}}(\boldsymbol{x}) f_{v,\mathcal{B}}(\boldsymbol{x}). \tag{48}$$

Thus we have proven that $\mathcal{L}_f(\mathcal{D}, \mathcal{B}, \mathcal{P}) = \mathcal{L}_f(\mathcal{D}, \mathcal{B}, \mathcal{P}')$.

**Property 3**    When features are independent, the Marginal Decomposition falls back to the ANOVA decomposition (see Appendix A.2). In this case, the functional components become zero-mean, uncorrelated, and have variance $\sigma_u^2$. Using Equation 16 and setting $\mathcal{B}_{\text{ind}} = \mathcal{D}$

$$\mathcal{L}_f(\mathcal{B}_{\text{ind}}, \mathcal{B}_{\text{ind}}, \mathcal{P}) = \sum_{u \subseteq [d]:|\mathcal{P}(u)| \geq 2} w(\mathcal{P}(u), \mathcal{P}(u)) \sigma_u^2 \tag{49}$$

and

$$\mathcal{L}_f(\mathcal{B}_{\text{ind}}, \mathcal{B}_{\text{ind}}, \mathcal{P}') = \sum_{u \subseteq [d]:|\mathcal{P}'(u)| \geq 2} w(\mathcal{P}'(u), \mathcal{P}'(u)) \sigma_u^2.$$

We define the sets

$$S_{\mathcal{P}} = \{u \subseteq [d] : |\mathcal{P}(u)| \geq 2\} \text{ and } S_{\mathcal{P}'} = \{u \subseteq [d] : |\mathcal{P}'(u)| \geq 2\}.$$

By Lemma B.4, we have that $S_{\mathcal{P}'} \subseteq S_{\mathcal{P}}$ and so

$$\begin{aligned}
\mathcal{L}_f(\mathcal{B}_{\text{ind}}, \mathcal{B}_{\text{ind}}, \mathcal{P}) - \mathcal{L}_f(\mathcal{B}_{\text{ind}}, \mathcal{B}_{\text{ind}}, \mathcal{P}') &= \sum_{u \in S_{\mathcal{P}}} w(\mathcal{P}(u), \mathcal{P}(u)) \sigma_u^2 - \sum_{u \in S_{\mathcal{P}'}} w(\mathcal{P}'(u), \mathcal{P}'(u)) \sigma_u^2 \\
&= \sum_{u \in S_{\mathcal{P}} \setminus S_{\mathcal{P}'}} w(\mathcal{P}(u), \mathcal{P}(u)) \sigma_u^2 \\
&\quad + \sum_{u \in S_{\mathcal{P}'}} [w(\mathcal{P}(u), \mathcal{P}(u)) - w(\mathcal{P}'(u), \mathcal{P}'(u))] \sigma_u^2 \\
&\geq 0 \qquad (\text{Since } w(\mathcal{P}(u), \mathcal{P}(u)) \geq w(\mathcal{P}'(u), \mathcal{P}'(u)) \geq 0)
\end{aligned}$$

$\square$

The following Theorem highlights a sufficient condition to define partition loss functions using explanation disagreements.

**Theorem B.4** (**Theorem 3.3**). *The $L_2$ disagreements $D_{L_2}(\phi^{Occ}, \phi')$ between Occlusion and the Arch/LIME/RISE/SHAP explainers respect Definition 3.1.*

*Proof.* We demonstrate that the $L_2$ disagreements between the post-hoc explanation methods respect the premise of Lemma B.5.

$$D_{L_2}(\boldsymbol{\phi}, \boldsymbol{\phi}') = \underset{\boldsymbol{x} \sim \mathcal{D}}{\mathbb{E}} \Bigg[ \sum_{i=1}^{D} \Bigg( \sum_{u \subseteq [d]: i \in \mathcal{P}(u), |\mathcal{P}(u)| \geq 2} \underbrace{(h(|\mathcal{P}(u)|) - h'(|\mathcal{P}(u)|))}_{b(|\mathcal{P}(u)|)} f_{u,\mathcal{B}}(\boldsymbol{x}) \Bigg)^2 \Bigg]$$

$$= \underset{\boldsymbol{x} \sim \mathcal{D}}{\mathbb{E}} \Bigg[ \sum_{i=1}^{D} \sum_{\substack{u,v \subseteq [d] \\ i \in \mathcal{P}(u), |\mathcal{P}(u)| \geq 2 \\ i \in \mathcal{P}(v), |\mathcal{P}(v)| \geq 2}} b(|\mathcal{P}(u)|) b(|\mathcal{P}(v)|) f_{u,\mathcal{B}}(\boldsymbol{x}) f_{v,\mathcal{B}}(\boldsymbol{x}) \Bigg]$$

$$= \underset{\boldsymbol{x} \sim \mathcal{D}}{\mathbb{E}} \Bigg[ \sum_{u,v \subseteq [d]: |\mathcal{P}(u)| \geq 2, |\mathcal{P}(v)| \geq 2} |\mathcal{P}(u) \cap \mathcal{P}(v)| \, b(|\mathcal{P}(u)|) b(|\mathcal{P}(v)|) f_{u,\mathcal{B}}(\boldsymbol{x}) \, f_{v,\mathcal{B}}(\boldsymbol{x}) \Bigg].$$

The corresponding interaction penalization is $w(\mathcal{P}(u), \mathcal{P}(v)) := |\mathcal{P}(u) \cap \mathcal{P}(v)| \, b(|\mathcal{P}(u)|) b(|\mathcal{P}(v)|)$. Now, obviously $w(\mathcal{P}(u), \mathcal{P}(u)) = |\mathcal{P}(u)| b(|\mathcal{P}(u)|)^2 \geq 0$ but $w(\mathcal{P}(u), \mathcal{P}(u)) \geq w(\mathcal{P}'(u), \mathcal{P}'(u))$ only holds for any superpartition $\mathcal{P}'$ if we compare certain pairs of explainers : Occ-Arch, Occ-SHAP, Occ-LIME, Occ-RISE. $\qquad \square$

While the Theorem highlights a sufficient condition for explanation disagreements to respect Definition 3.1, one might wonder whether these conditions are also necessary. The following result partially answers that question by demonstrating that alternative measures of disagreements do not respect Definition 3.1.

**Theorem B.5** (**Theorem 3.4**). *The $L_2$ disagreements between explanation pairings that do not include Occlusion (e.g. LIME vs SHAP) break property 3 of Definition 3.1. Moreover, using the Pearson correlation disagreement $D_\theta$ breaks property 2.*

*Proof.* To prove the two statements of the Theorem, we simply need to come up with a single counterexample where the measures of disagreement break one of the properties of Definition 3.1.

**(Counterexample for $L_2$ distance)** Let $f(\boldsymbol{x}) = \prod_{i=1}^{d} x_i$ be the model and let $\mathcal{B} = \mathcal{D} = U([-1,1])^d$ be a distribution with independent features. We show that $L_2$ disagreements that do not involve Occlusion break property 3. First, we have that $f_{u,\mathcal{B}}(\boldsymbol{x}) = 0$ for any $u \neq [d]$ and $f_{[d],\mathcal{B}}(\boldsymbol{x}) = \prod_{i=1}^{d} x_i$. If we group the features into $D$ groups, by Theorem 3.1,

$$\phi_i^\mu(f, \boldsymbol{x}, \mathcal{B}, \mathcal{P}) = h(D) f_{[d],\mathcal{B}}(\boldsymbol{x}). \tag{50}$$

Recall that $h(D) = 0$ for ArchAttribute, $h(D) = 1/D$ for SHAP and $h(D) = 1/2^{D-1}$ for LIME/RISE. Thus, we have

$$D_{L_2}(\boldsymbol{\phi}, \boldsymbol{\phi}') = \underset{\boldsymbol{x} \sim \mathcal{D}}{\mathbb{E}} \Bigg[ \sum_{i=1}^{D} \Big( (h(D) - h'(D)) f_{[d],\mathcal{B}}(\boldsymbol{x}) \Big)^2 \Bigg]$$

$$= D(h(D) - h'(D))^2 \underset{\boldsymbol{x} \sim \mathcal{D}}{\mathbb{E}} \big[ f_{[d],\mathcal{B}}(\boldsymbol{x})^2 \big]$$

$$= D(h(D) - h'(D))^2 \underset{\boldsymbol{x} \sim \mathcal{D}}{\mathbb{E}} \Bigg[ \prod_{i=1}^{d} x_i^2 \Bigg]$$

$$= D \frac{(h(D) - h'(D))^2}{3^d}.$$

Since all variables are symmetric in this toy example, explanation disagreements depend only on the number of feature groups $D$ and not on the groups themselves. Now, property 3 of Definition 3.1 implies that this function should not increase as $D$ is reduced (*i.e.* as we fuse groups together). However, this is not true when comparing Arch-LIME, Arch-SHAP, and LIME-SHAP.

**(Counterexample for $D_\theta$)** Let $f(\boldsymbol{x}) = x_1 + x_2 + x_3 x_4$ be the model whose prediction at $\boldsymbol{x} = \boldsymbol{1}$ is compared to the one at $\boldsymbol{z} = \boldsymbol{0}$. When each feature is considered as a separate group, the post-hoc explanations take the form $\boldsymbol{\phi}(f, \boldsymbol{x}, \boldsymbol{z}, \mathcal{P}) = [1, 1, h(2), h(2)]$. Note that RISE/LIME and SHAP will yield the same explanation since the largest interaction is of order 2. By rephrasing Equation 9 as an agreement metric (instead of disagreement), and setting $\mathcal{D} = \delta_{\boldsymbol{x}}$ and $\mathcal{B} = \delta_{\boldsymbol{z}}$ as the distributions, we get

$$\frac{\langle \boldsymbol{\phi}, \boldsymbol{\phi}' \rangle}{\|\boldsymbol{\phi}\|\|\boldsymbol{\phi}'\|} = \frac{2 + 2h(2)h'(2)}{\sqrt{2 + 2h(2)^2}\sqrt{2 + 2h'(2)^2)}} = \frac{1 + h(2)h'(2)}{\sqrt{1 + h(2)^2}\sqrt{1 + h'(2)^2}}. \tag{51}$$

Now, since the model is additive in features $x_1$ and $x_2$, an ideal loss function should remain unchanged whenever these features are fused into a single group. Otherwise, reducing the loss does not guarantee that we have found relevant interactions between the features. If we define a new partition $\mathcal{P}'$ where the first two features are fused, we get the explanations $\boldsymbol{\phi}(f, \boldsymbol{x}, \boldsymbol{z}, \mathcal{P}') = [2, h(2), h(2)]$. The corresponding cosine agreements are

$$\frac{\langle \boldsymbol{\phi}, \boldsymbol{\phi}' \rangle}{\|\boldsymbol{\phi}\|\|\boldsymbol{\phi}'\|} = \frac{4 + 2h(2)h'(2)}{\sqrt{4 + 2h(2)^2}\sqrt{4 + 2h'(2)^2)}} = \frac{2 + h(2)h'(2)}{\sqrt{2 + h(2)^2}\sqrt{2 + h'(2)^2}}. \tag{52}$$

For any pairing between Arch ($h(2) = 0$), Occlusion ($h(2) = 1$), and SHAP/LIME/RISE ($h(2) = 1/2$)), Equation 52 is larger than Equation 51. As a result, agreement can be increased between explanation methods without infering meaningful groups.

$\square$

## C  AGREED ALGORITHM

The AGREED algorithm aims at minimizing the $L_2$ disagreements between the joint-PDP/ArchAttribute and joint-PFI/Occlusion explanations

$$\mathcal{L}_f^{\text{AGREED}}(\mathcal{D}, \mathcal{B}, \mathcal{P}) := \sum_{i=1}^{D} \Psi(i) \quad \text{with} \quad \Psi(i) := \underset{\boldsymbol{x} \sim \mathcal{D}}{\mathbb{E}} \left[ (\phi_i^{\text{Arch}}(f, \boldsymbol{x}, \mathcal{B}, \mathcal{P}) - \phi_i^{\text{Occ}}(f, \boldsymbol{x}, \mathcal{B}, \mathcal{P}))^2 \right].$$
(53)

Starting from a granular partition $\{\{1\}, \dots, \{d\}\}$, we greedily minimize Equation 53

1. Select the group $i$ with highest potential $\Psi(i)$.
2. Compute its pairwise interaction with other groups $j$.
3. Fuse group $i$ with the group $j$ of maximal pairwise interaction.
4. Repeat until the disagreements fall below $\epsilon$.

The implementation details of the algorithm depend on whether $\mathcal{B} = \mathcal{D}$ or not.

### C.1  CASES WHERE $\mathcal{B} = \mathcal{D}$

It is common to set $\mathcal{B} = \mathcal{D}$ on Tabular data so that $\boldsymbol{x}$ and $\boldsymbol{b}$ can both be interpreted as a random sample from the dataset. This assumption introduces a symmetry between $\boldsymbol{x}$ and $\boldsymbol{b}$ that makes statistical estimates more efficient.

**Step 1.  Computing the Group Potential**  The building blocks of AGREED are the following $D \times N \times N$ tensors

**Definition C.1.** *Let $\mathcal{P}$ be a partition of $[d]$ into $D$ disjoint groups, $\mathcal{D}$ be the data distribution and $\{\boldsymbol{x}^{(k)}\}_{k=1}^N$ be $N$ points sampled from it. We define the $D \times N \times N$ tensor $\boldsymbol{G}$*

$$G_{i,k,\ell} = f(\boldsymbol{r}_{\mathcal{P}^{-1}(\{i\})}(\boldsymbol{x}^{(\ell)}, \boldsymbol{x}^{(k)})) - f(\boldsymbol{x}^{(\ell)}).$$
(54)

These tensors are of interest because averaging them along their second and third axis leads to consistent estimate of the joint-PDP/ArchAttribute and joint-PFI/Occlusion respectively

$$\frac{1}{N} \sum_{\ell=1}^{N} G_{i,k,\ell} \xrightarrow{p} \phi_i^{\text{Arch}}(f, \boldsymbol{x}^{(k)}, \mathcal{D}, \mathcal{P})$$
(55)

$$-\frac{1}{N} \sum_{k=1}^{N} G_{i,k,\ell} \xrightarrow{p} \phi_i^{\text{Occ}}(f, \boldsymbol{x}^{(\ell)}, \mathcal{D}, \mathcal{P}).$$
(56)

Given $\boldsymbol{G}$, we can efficiently compute the potential of group $i$

$$\frac{1}{N} \sum_{k=1}^{N} \left( \frac{1}{N} \sum_{\ell=1}^{N} (G_{i,k,\ell} + G_{i,\ell,k}) \right)^2 \xrightarrow{p} \Psi(i).$$
(57)

**2. Computing Interaction Between Groups**  At any point in the iterative algorithm, we will have access to the $\boldsymbol{G}$ tensor of the current partition. Thus, a good between-group interaction score should leverage this precomputed tensor to avoid unnecessary model inferences.

**Definition C.2.** *Let $\mathcal{P}$ be a partition of $[d]$ into $D$ disjoint groups, $i \in [D]$ be a group that we want to extend, $\mathcal{D}$ be the data distribution, and $\{\boldsymbol{x}^{(k)}\}_{k=1}^N$ be $N$ points sampled from it. We define the $(D-1) \times N \times N$ matrix $\boldsymbol{I}$ such that*

$$I_{j,k,\ell} := f(\boldsymbol{r}_{\mathcal{P}^{-1}(\{i,j\})}(\boldsymbol{x}^{(\ell)}, \boldsymbol{x}^{(k)})) - f(\boldsymbol{r}_{\mathcal{P}^{-1}(\{i\})}(\boldsymbol{x}^{(\ell)}, \boldsymbol{x}^{(k)})) - f(\boldsymbol{r}_{\mathcal{P}^{-1}(\{j\})}(\boldsymbol{x}^{(\ell)}, \boldsymbol{x}^{(k)})) + f(\boldsymbol{x}^{(\ell)}).$$
(58)

Crucially, $\boldsymbol{I}$ can be computed efficiently by querying $\boldsymbol{G}$

$$I_{j,k,\ell} = f(\boldsymbol{r}_{\mathcal{P}^{-1}(\{i,j\})}(\boldsymbol{x}^{(\ell)}, \boldsymbol{x}^{(k)})) - f(\boldsymbol{x}^{(\ell)}) - G_{i,k,\ell} - G_{j,k,\ell}. \tag{59}$$

Averaging this tensor along the third axis yields

$$\frac{1}{N}\sum_{\ell=1}^{N} I_{j,k,\ell} \xrightarrow{p} \sum_{u\subseteq[d]:\mathcal{P}(u)=\{i,j\}} f_{u,\mathcal{D}}(\boldsymbol{x}^{(k)}), \tag{60}$$

a pure measure of interactions only involving features from groups $i$ and $j$. Averaging the tensor along its second axis leads to

$$\frac{1}{N}\sum_{k=1}^{N} I_{j,k,\ell} \xrightarrow{p} \sum_{u\subseteq[d]:\{i,j\}\subseteq\mathcal{P}(u)} f_{u,\mathcal{D}}(\boldsymbol{x}^{(\ell)}), \tag{61}$$

a full measure of interactions involving features from groups $i$, $j$, and possibly other groups. To compute the strength of the interaction between two groups $i$ and $j$, we could report the pure interaction, the full interaction, or a weighted average. Like (Tsang et al., 2020), we average the pure and full interactions with weights $0.5$.

**3. Fusing Groups** After having identified two groups $i$ and $j$ that interact, we define a superpartition $\mathcal{P}'$ of size $D - 1$ where $i, j$ are fused into a single group. The $D - 2$ groups $k \neq i, j$ are simply re-indexed from 1 to $D - 2$, while $i$ and $j$ are considered the $(D - 1)^{\text{th}}$ group. Since the $\boldsymbol{G}$ tensor is relative to the current partition, it must be updated to $\boldsymbol{G}'$ when performing group fusion. For the $D - 2$ groups that were not fused, we copy their $G_{k,:,:}$ values. For the 2 groups that were fused, we store the joint effect

$$G'_{D-1,k,\ell} := f(\boldsymbol{r}_{\mathcal{P}^{-1}(\{i,j\})}(\boldsymbol{x}^{(\ell)}, \boldsymbol{x}^{(k)})) - f(\boldsymbol{x}^{(\ell)}) = I_{j,k,\ell} + G_{i,k,\ell} + G_{j,k,\ell}, \tag{62}$$

that is computed without additional model inference. Here is the pseudocode for updating the partition.

---

**Algorithm 1** Update Partition $\mathcal{P}$ by fusing $i, j \in [D]$ into a new group

---

1: **procedure** UPDATE_PARTITION($\mathcal{P}, \boldsymbol{G}, \boldsymbol{I}, i, j$)
2:      Initialize new partition map $\mathcal{P}'$;
3:      $G' \leftarrow \texttt{zeros}(D - 1, N, N)$;
4:      % Count Variable
5:      $c \leftarrow 1$
6:      **for** $k \in [D]$ such that $k \neq i, j$ **do**
7:          Define $\mathcal{P}'(\ell) := c, \quad \forall \ell \in \mathcal{P}^{-1}(\{k\})$;
8:          $G'_{c,:,:} \leftarrow G_{k,:,:}$;
9:          $c \leftarrow c + 1$;
10:      % Groups $i$ and $j$ are fused
11:      Define $\mathcal{P}'(\ell) := D - 1, \quad \forall \ell \in \mathcal{P}^{-1}(\{i,j\})$;
12:      $G'_{D-1,:,:} \leftarrow I_{j,:,:} + G_{i,:,:} + G_{j,:,:}$;
13:      **return** $\mathcal{P}', \boldsymbol{G}'$

---

This partition update requires no model inference since all relevant computations are queried from the $\boldsymbol{G}$ and $\boldsymbol{I}$ tensors. Putting everything togheter, we end up with Algorithm 2. AGREED requires $\mathcal{O}(d^2 N^2)$ model inferences since line 5 calls $f$ $dN^2$ times, then for each iteration of the while loop (of which there are at most $d - 1$), $\boldsymbol{I}$ is computed which does not call $f$ more than $dN^2$ times.

## C.2 CASES WHERE $\mathcal{B} \neq \mathcal{D}$

When computing saliency maps for Image Classification, it is common to use a single baseline $\mathcal{B} = \delta_{\boldsymbol{b}}$ (typically an image with no information). Also, there is no need to find a partition $\mathcal{P}$ that works across all images $\boldsymbol{x}$ from the dataset. Finding a partition that works on a single image $\boldsymbol{x}$ is a more realistic goal so we set $\mathcal{D} = \delta_{\boldsymbol{x}}$.

---

**Algorithm 2** Adaptive Grouping to ReducE Explanation Disagreements (requires $\mathcal{D} = \mathcal{B}$).

---

1: **procedure** AGREED($f, \{\boldsymbol{x}^{(k)}\}_{k=1}^{N} \sim \mathcal{D}^{N}, \epsilon$)
2:     % Initialization
3:     Initialize partition $\mathcal{P}$ such that $\mathcal{P}(j) := j \ \ \forall j \in [d]$;
4:     $D \leftarrow d$;
5:     Compute the $D \times N \times N$ tensor $\boldsymbol{G}$ (cf. Equation 54);
6:     Compute the potentials $\Psi(i), \ i = 1, 2, \ldots, D$ from $\boldsymbol{G}$ (cf. Equation 57);
7:     Obj $\leftarrow \sum_{i=1}^{D} \Psi(i)$;
8:     **while** Obj $> \epsilon$ **do**
9:         % Which group to extend
10:         $i \leftarrow \text{argmax}_{i=1,2,\ldots,D} \ \Psi(i)$;
11:         $\boldsymbol{I} \leftarrow \texttt{zeros}(D-1, N, N)$
12:         % Find the fuse candidate
13:         **for** $j \in [D]$ such that $j \neq i$ **do**
14:             Compute Between-Group Interaction $I_{j,:,:}$ (cf. Equation 59);
15:         $j = \text{argmax}_{j=1,2,\ldots,D-1} \frac{1}{2N} \sum_{k=1}^{N} \left[ \left( \frac{1}{N} \sum_{\ell=1}^{N} I_{j,k,\ell} \right)^2 + \left( \frac{1}{N} \sum_{\ell=1}^{N} I_{j,\ell,k} \right)^2 \right]$;
16:         % Fuse groups $i$ and $j$ leading to a superpartition
17:         $\mathcal{P}, \boldsymbol{G} \leftarrow$ UPDATE_PARTITION($\mathcal{P}, \boldsymbol{G}, \boldsymbol{I}, i, j$)
18:         $D \leftarrow D - 1$
19:         Compute the potentials $\Psi(i), \ i = 1, 2, \ldots, D$ from $\boldsymbol{G}$ (cf. Equation 57);
20:         Obj $\leftarrow \sum_{i=1}^{D} \Psi(i)$;
21:     **return** $\mathcal{P}, \boldsymbol{G}$;

---

Since AGREED expects $\mathcal{B} = \mathcal{D}$, we introduce a mixture distribution $\mathcal{Q} = \frac{1}{2}(\delta_{\boldsymbol{x}} + \delta_{\boldsymbol{b}})$ and note that

$$\mathcal{L}_f^{\text{AGREED}}(\delta_{\boldsymbol{x}}, \delta_{\boldsymbol{b}}, \mathcal{P}) = 4 \times \mathcal{L}_f^{\text{AGREED}}(\mathcal{Q}, \mathcal{Q}, \mathcal{P}). \tag{63}$$

Thus, we can apply AGREED on images by feeding Algorithm 2 with two samples ($\boldsymbol{x}$ and $\boldsymbol{b}$) from a fictional data distribution $\mathcal{Q}$. This will lead to a $D \times 2 \times 2$ $\boldsymbol{G}$ tensor and a $D - 1 \times 2 \times 2$ $\boldsymbol{I}$ tensor.

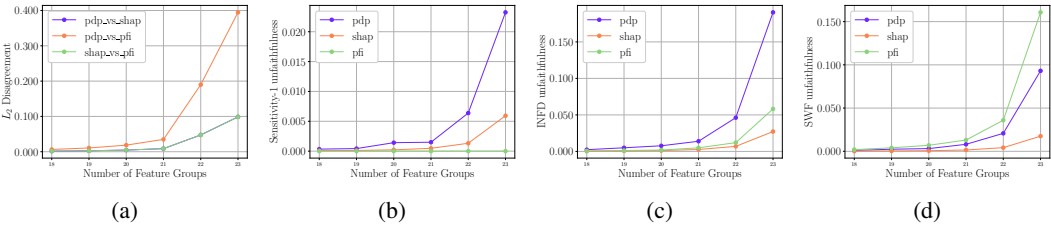

Figure 7: Tabular Data: EBM fitted on default credit. (a) Explanation $L_2$ Disagreement are reduced as AGREED groups more features. (b-c-d) Three inconsistent unfaithfulness metrics collectively converge to $0$ as AGREED groups features.

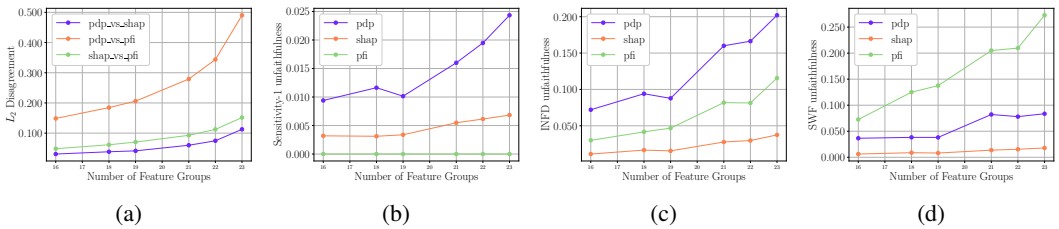

Figure 8: Tabular Data: GBT fitted on default credit. (a) Explanation $L_2$ Disagreement are reduced as AGREED groups more features. (b-c-d) Three inconsistent unfaithfulness metrics collectively converge to $0$ as AGREED groups features.

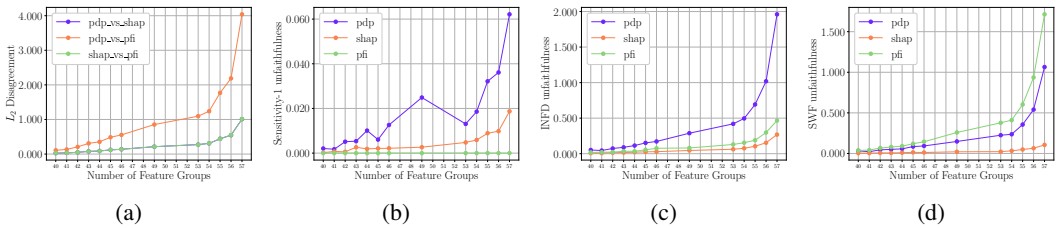

Figure 9: Tabular Data: EBM fitted on SPAM. (a) Explanation $L_2$ Disagreement are reduced as AGREED groups more features. (b-c-d) Three inconsistent unfaithfulness metrics collectively converge to $0$ as AGREED groups features.

## D EXTENDED EXPERIMENTS

### D.1 TABULAR TOY DATA

Tabular synthetic datasets are generated by first sampling $N$ samples of $d$ features $\boldsymbol{x} \sim \mathcal{N}(\boldsymbol{0}, \boldsymbol{B})$ with a block-diagonal covariance matrix. The feature groups forming each block are chosen randomly. Then, another random feature partition $\mathcal{P}$ is generated to define a group-wise additive model. Each component $g_{\mathcal{P}^{-1}(\{i\})}$ of the model is generated from the following list

1. $g_{\mathcal{P}^{-1}(\{i\})}(\boldsymbol{x}) = \prod_{j \in \mathcal{P}^{-1}(\{i\})} \prod_{k \in \mathcal{P}^{-1}(\{i\})} x_j x_k$
2. $g_{\mathcal{P}^{-1}(\{i\})}(\boldsymbol{x}) = \exp\left[-0.5 \sum_{j \in \mathcal{P}^{-1}(\{i\})} x_j^2\right]$
3. $g_{\mathcal{P}^{-1}(\{i\})}(\boldsymbol{x}) = \sigma(\sum_{j \in \mathcal{P}^{-1}(\{i\})} \omega_j x_j^2)$ where $\sigma$ is a Sine, Cosine, Tanh, ReLU.

### D.2 TABULAR DATA

#### D.2.1 ADDITIONAL QUANTITATIVE RESULTS

Figures 7-11 show the trade-offs between explanation disagreement/unfaithfulness and feature group sizes for EBM and GBT models fitted on Default-Credit, Spam, and NOMAO. The insights identical

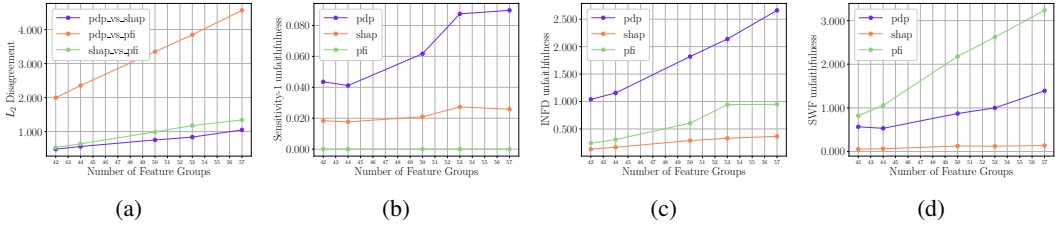

Figure 10: Tabular Data: GBT fitted on Spam. (a) Explanation $L_2$ Disagreement are reduced as AGREED groups more features. (b-c-d) Three inconsistent unfaithfulness metrics collectively converge to $0$ as AGREED groups features.

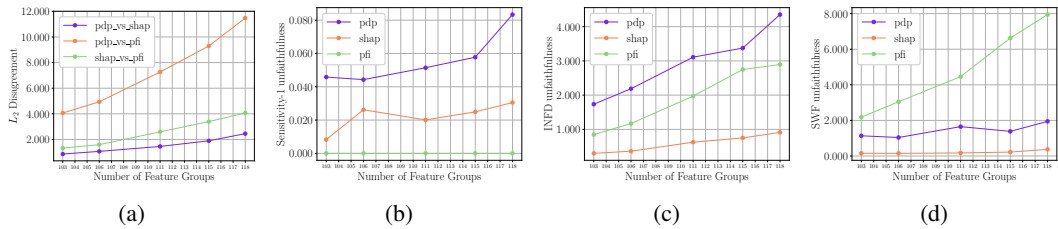

Figure 11: Tabular Data: GBT fitted on Nomao. (a) Explanation $L_2$ Disagreement are reduced as AGREED groups more features. (b-c-d) Three inconsistent unfaithfulness metrics collectively converge to $0$ as AGREED groups features.

to those discussed in the main manuscript: as we group features together, disagreements between PDP/SHAP/PFI are reduced and the Sensitivity-1, INFD, SWF unfaithfulness metrics also collectively converge to zero.

The positive effects of feature grouping are most prominent on EBMs compared to GBTs. This is because EBM are restricted to only model interaction of order 2, while GBTs with depth-$T$ trees can model interaction whose order at-most $T$. Apparently, the GBTs trained on the two largest datasets (Figures 10 & 11) have learned very high-order interactions that are extremely hard to minimize. Although AGREED fails to find agreement in those two settings, the algorithm is still useful to warn the user that the model might be too complicated to be explained with feature-based explanations. Hence, it might be best to rely on a EBM if faithful and unambiguous explanations are desired.

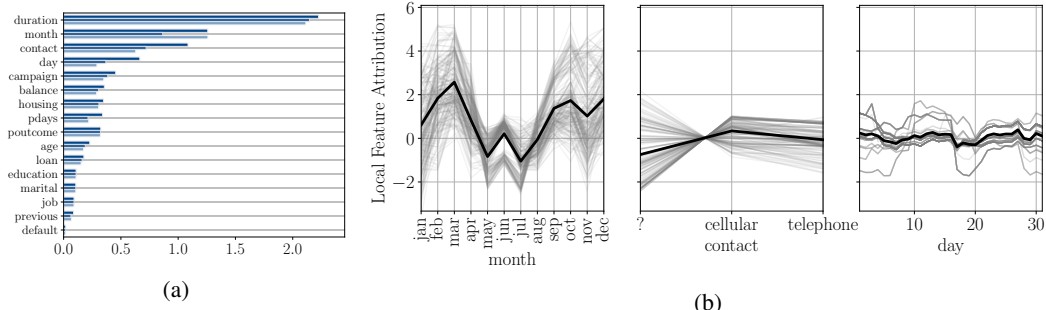

Figure 12: Marketing with no grouping. a) The global feature importance according to the PFI (opaque), SHAP (semi-transparent), and PDP (transparent) explainers. b) The PDPs of three disagreeing features (thick black line) along with the ICE (thin lines).

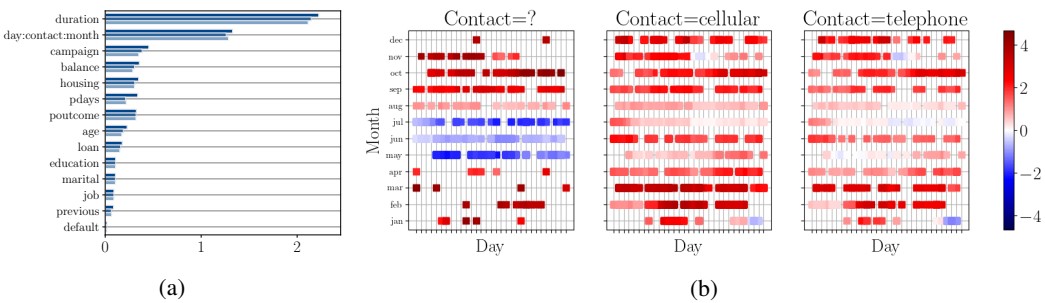

Figure 13: Marketing with grouping. a) The global feature importance according to the PFI (opaque), SHAP (semi-transparent), and PDP (transparent) explainers. b) The joint-PDP of the `month:day:contact` feature group.

### D.2.2 QUALITATIVE RESULTS

The challenge in interpreting joint-importance scores is that, when features $j$, $k$ and $\ell$ are treated as a group $i$, their joint-PDP $\phi_i^{\text{PDP}}(h, \boldsymbol{x}, \mathcal{B}, \mathcal{P})$ becomes a multivariate function of $x_j$, $x_k$ and $x_\ell$. Humans are notoriously bad at visualizing high-dimensional functions, so we advocate selecting 1-2 features to plot while *conditioning* the remaining ones. When features are binary or categorical, condition on their unique values. For numerical features, condition on their quantiles.

**Marketing** The Marketing dataset describes the marketing campaign of a Portuguese banking institution. Each instance corresponds to a distinct phone call and the binary label encodes whether the client subscribed to a term deposit. We explain an EBM fitted on this dataset using the PDP/SHAP/PFI explainers. From Figure 12 (a), the three techniques attribute very different global importances to the features `month`, `day`, and `contact`. Figure 12 (b) shows the PDP of these three features along side their Individual Conditional Expectation (ICE) (Goldstein et al., 2015). The ICE curves can be interpreted as the local trend when varying $x_j$ while the PDP is the average trend. It is apparent that the average trend is very different from local ones, especially at the value `contact=?`. These disagreements are induced by strong feature interactions within the model.

To reduce the disagreements caused by feature interactions, we ran AGREED using $N = 1000$ random samples for MC estimates and $\epsilon = 0.25$ as the stopping criterion (it stops when the loss reduces to 25% of its original value). In 30 seconds, AGREED discovered the group `month:day:contact`. According to Figure 13 (a), there are now almost no disagreements between the global group importance reported by PDP/SHAP/PFI. The joint-PDP of `month:day:contact` can be visualized using a scatter plot along `month:day` while conditioning on different values of `contact` $\in$ [?, `cellular`, `telephone`]. This is presented in Figure 13 (b). When `contact=?`, there is a significant drop in model output during June and July compared to other values of `contact`. Moreover, the trends along `month` and `day` hardly appear to be additive : there is a sharp drop in late January that does not occur in other months. Therefore, it is better to interpret them jointly as a single "date" feature group.

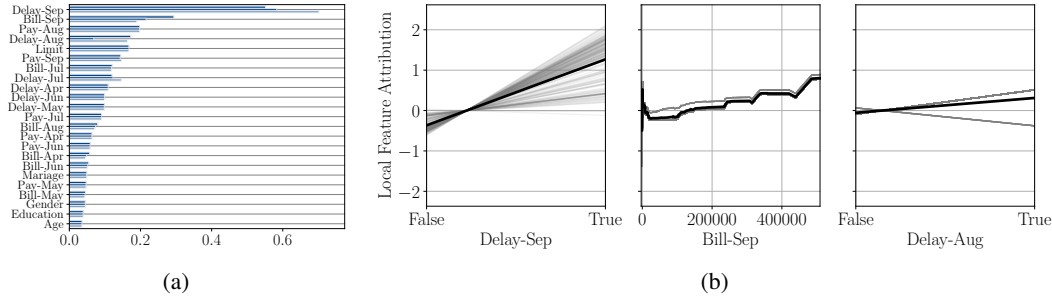

Figure 14: Default-Credit with no grouping. a) The global feature importance according to the PFI (opaque), SHAP (semi-transparent), and PDP (transparent) explainers. b) The PDPs of three important features (thick black line) along with the ICE (thin lines).

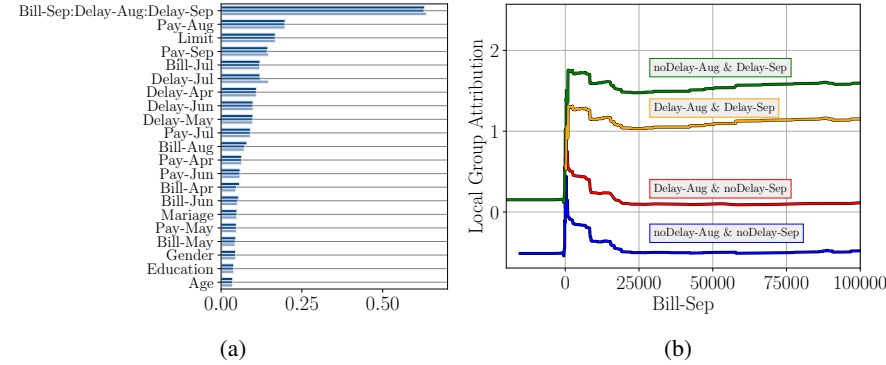

Figure 15: Default-Credit with grouping. a) The global feature importance according to the PFI (opaque), SHAP (semi-transparent), and PDP (transparent) explainers. b) The joint-PDP of the `Delay-Aug:Delay-Sep:Bill-Sep` feature group.

**Default-Credit**  The Default-Credit dataset aims at predicting if clients of a Taiwanese bank will default on their credit. The data contains records of 30K individuals and 23 features related to past payments/bills/delays and demographic characteristics. We explain an EBM fitted on this dataset using the PDP/SHAP/PFI explainers. Figure 14 (a) demonstrates that the three explainers provide different global importances to the features `Delay-Sep`, `Delay-Aug`, and `Bill-Sep`. Figure 14 (b) shows their PDP and ICE local attribution. By comparing the PDP and ICE, it clear that the PDP of `Delay-Aug` is misleading since having a delayed payment in August sometimes increases the model output and sometimes decreases it. This suggests that the effects of `Delay-Aug` cannot be faithfully described using single feature importance score.

To faithfully explain the effects of interacting features, we ran AGREED using $N = 1000$ random samples for MC estimates and $\epsilon = 0.1$ as the stopping criterion (AGREED stops when the loss reduces to $10\%$ of its original value). After about a minute, AGREED yielded the group `Delay-Aug:Delay-Sep:Bill-Sep`. According to Figure 15 (a), PDP/SHAP/PFI now agree on the global importance of each feature group. The joint-PDP of `Delay-Aug:Delay-Sep:Bill-Sep` is a multivariate function involving two binary features and a numerical one. This function can be visualized by plotting four line charts w.r.t `Bill-Sep` (one line for each configuration of the remaining two binary variables). See Figure 15 (b) for the results. Interestingly, the impact of August delays depends on whether there was a September delay. Comparing the yellow curve to the green one, and the red/blue curves, the effect of `Delay-Aug` is completely reversed depending on `Delay-Sep`. The effect of `Bill-Sep` also depends on `Delay-Sep`. We are not sure why these trends are happening in the data, but at least AGREED warns us that trends involving `Delay-Aug:Delay-Sep:Bill-Sep` are inherently high-dimensional and should be visualized accordingly.

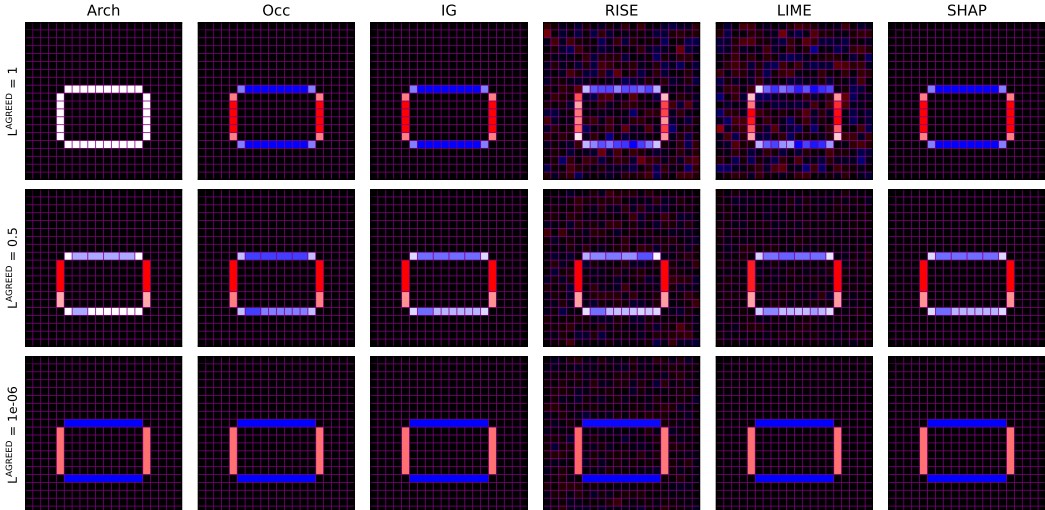

Figure 16: Example of applying AGREED on the toy rectangle classification dataset. The plots show the image overlaid with the patch importance. Blue and red refer to negative and positive attributions, respectively. As AGREED converges (larger patches are found), the various explainability methods increase in agreement.

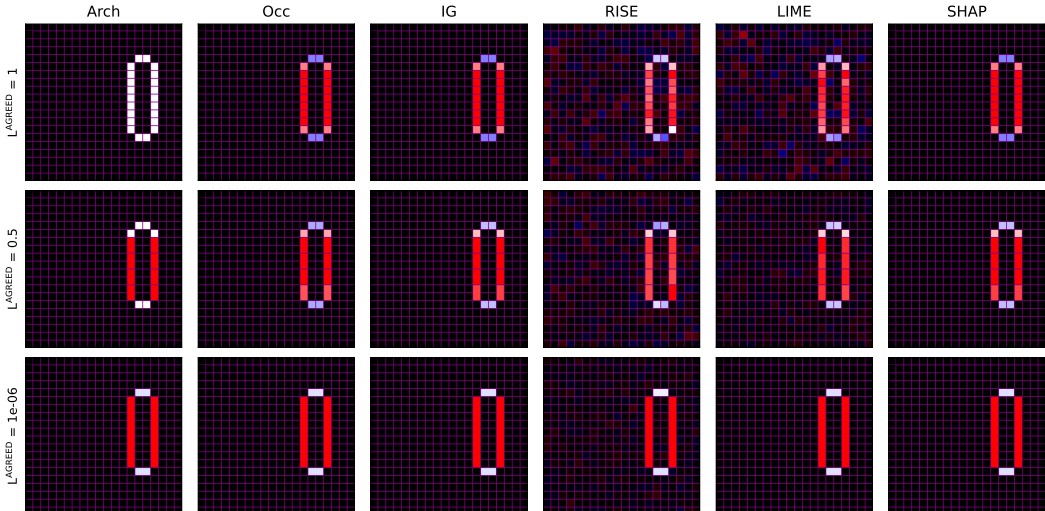

Figure 17: Another example of applying AGREED on the toy rectangle classification dataset. The plots show the image overlayed with the patch importance. Blue and red refer to negative and positive attributions respectivelly. As AGREED converges (larger patches are found), the various explainability methods increase in agreement.

### D.3   IMAGE TOY DATA

Figures 16 & 17 show qualitative results for the toy image dataset consisting of predicting whether rectangles are tall ($y = 1$) or wide ($y = 0$). Once each edge of the rectangle becomes its own group, all explainability methods agree on their importance.

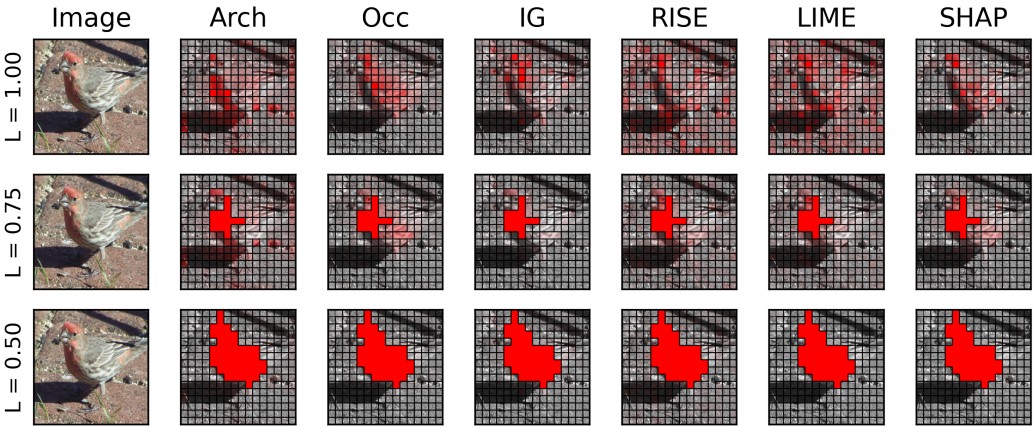

Figure 18: Explaining the "House Finch" prediction of a ResNet18. AGREED yields a partition with increased agreement between the various saliency map methods.

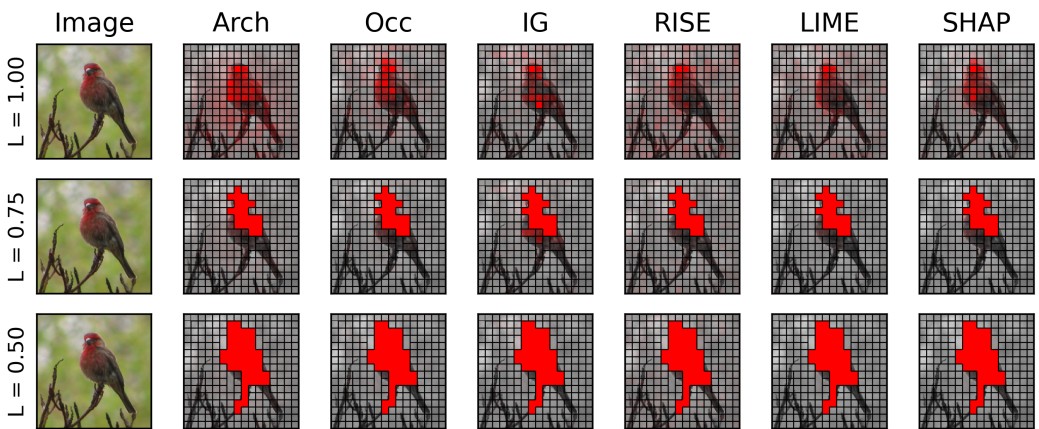

Figure 19: Explaining the "House Finch" prediction of a ResNet18. AGREED yields a partition with increased agreement between the various saliency map methods.

## D.4 MINIIMAGENET

Figures 18 to 27 present the saliency maps resulting from the AGREED partitions. The model under study is a ResNet18 pre-trained on ImageNet. We see that, in general, AGREED identifies a large patch of great importance that covers the animal. However, there are exceptions: in Figures 20 & 23 various patches cover specific parts of the animal.

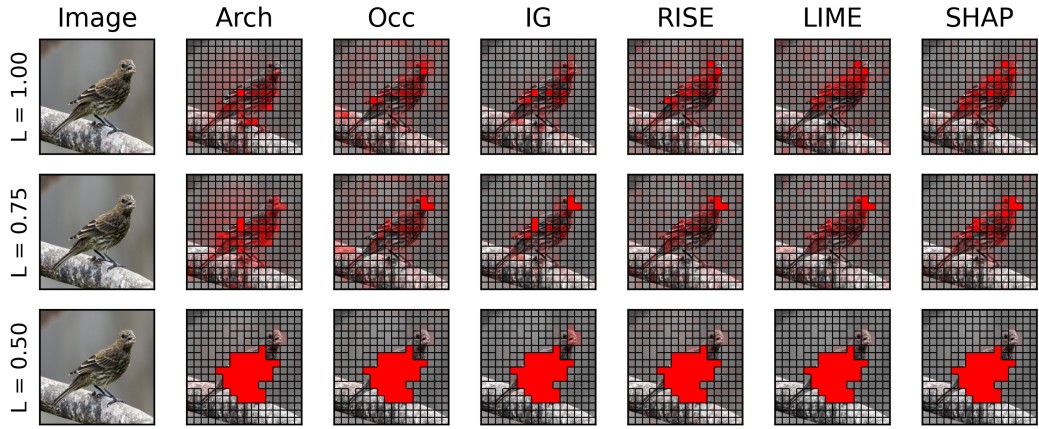

Figure 20: Explaining the "House Finch" prediction of a ResNet18. AGREED yields a partition with increased agreement between the various saliency map methods.

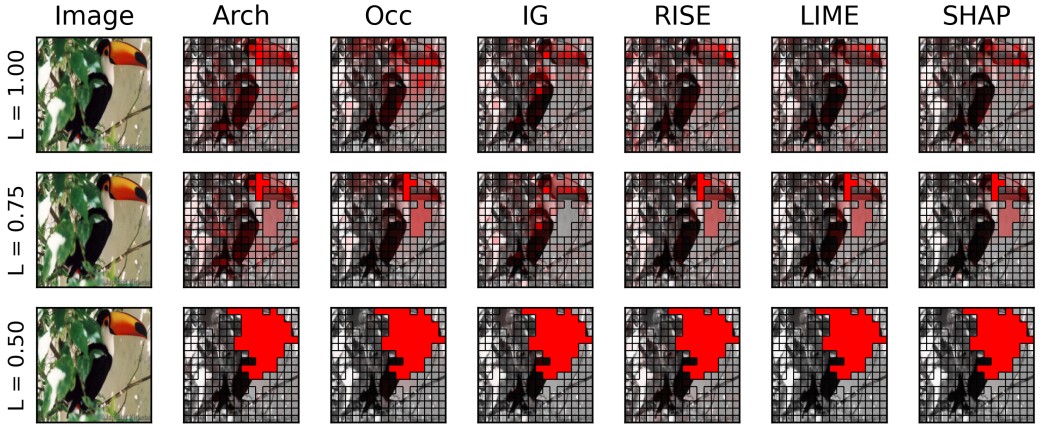

Figure 21: Explaining the "Toucan" prediction of a ResNet18. AGREED yields a partition with increased agreement between the various saliency map methods.

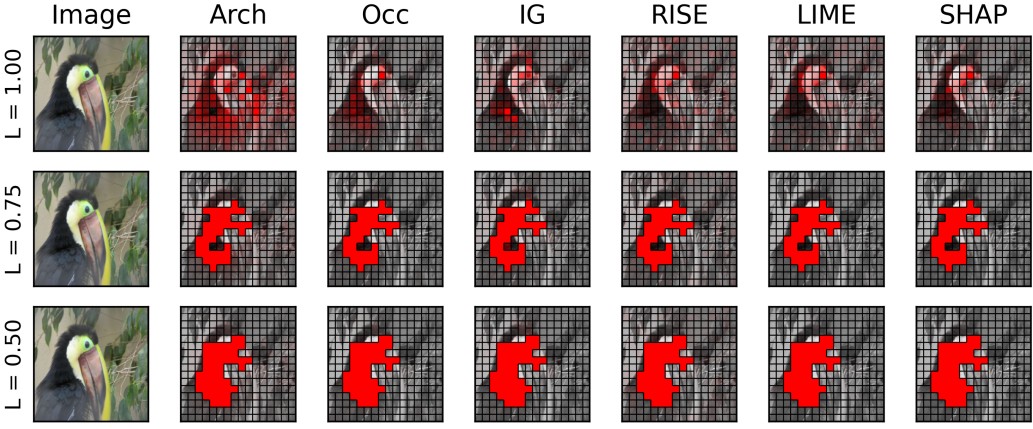

Figure 22: Explaining the "Toucan" prediction of a ResNet18. AGREED yields a partition with increased agreement between the various saliency map methods.

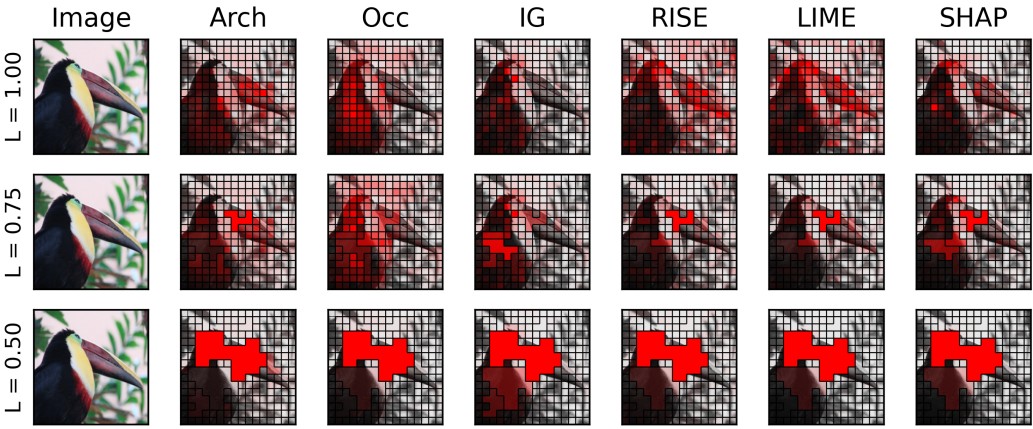

Figure 23: Explaining the "Toucan" prediction of a ResNet18. AGREED yields a partition with increased agreement between the various saliency map methods.

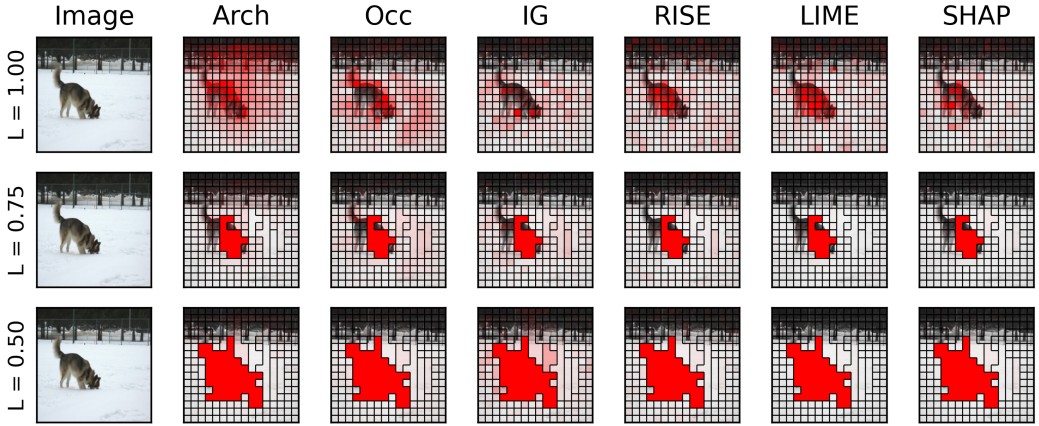

Figure 24: Explaining the "Malamute" prediction of a ResNet18. AGREED yields a partition with increased agreement between the various saliency map methods.

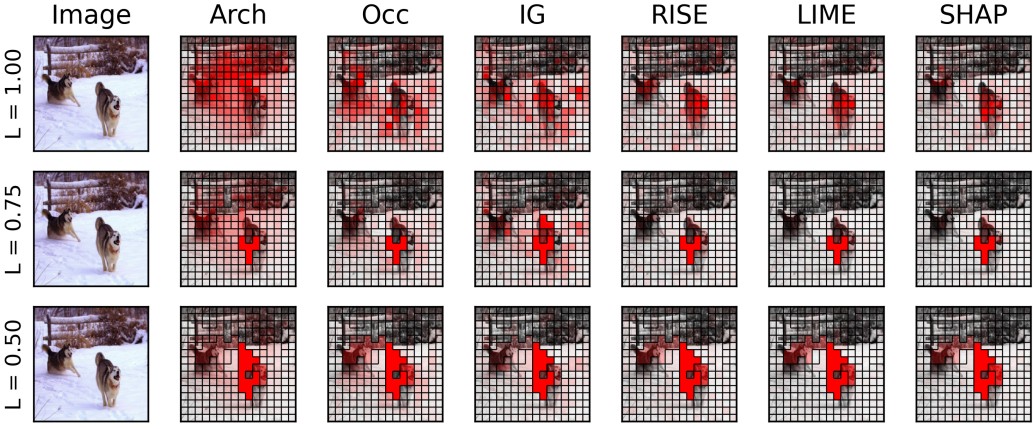

Figure 25: Explaining the "Malamute" prediction of a ResNet18. AGREED yields a partition with increased agreement between the various saliency map methods.

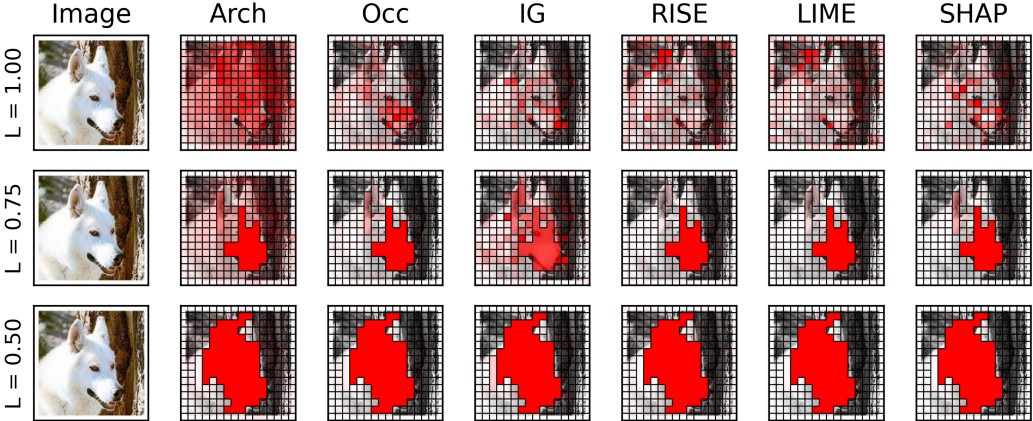

Figure 26: Explaining the "White Wolf" prediction of a ResNet18. AGREED yields a partition with increased agreement between the various saliency map methods.

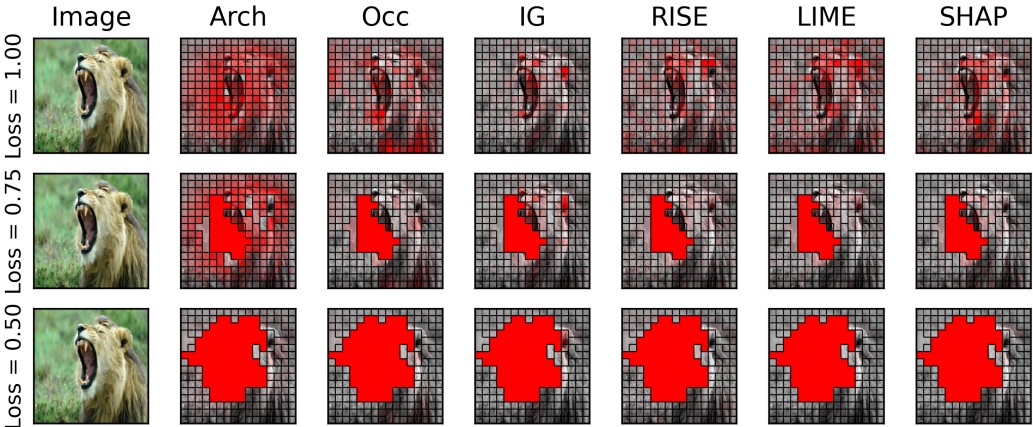

Figure 27: Explaining the "Lion" prediction of a ResNet18. AGREED yields a partition with increased agreement between the various saliency map methods.

