# OpenReview forum: "Tackling the XAI Disagreement Problem with Adaptive Feature Grouping"
_ICLR.cc/2026/Conference — ICLR 2026 Poster_

### Official Review · Reviewer_Dpg6 · 2025-10-31

**Soundness:** 3
**Presentation:** 2
**Contribution:** 3
**Rating:** 4
**Confidence:** 4

**Summary:**

The paper addresses disagreement problem in explainable ai, where different post-hoc explanation methods and metric often produce inconsistent results. The paper extend the framework of functional decomposition to analyze explanations over groups of features. It has the theoretical contribution to prove that the disagreement between various group-based explainers and high scores on common metrics share a common root cause. Therefore the paper proposes AGREED, an algorithm that adaptively finds feature partitions that minimize these interactions.

**Strengths:**

1. The paper is well written and easy to follow.
2. The paper has rigorous theoretical contribution.
3. The paper clearly demonstrates that both disagreements between explanation method and (un)faithfulness as measured by standard metric are functions of the same underlying between-group interaction terms.
4. The paper validated the theoretical claims in diverse modalities both synthetic and real world dataset.

**Weaknesses:**

1. The problem statement in the introduction is very broad. The described challenge is fundamental to almost all xai research.
2. The readability of the figures are poor, also the main text needs to provide guidance on how to interpret the results in the figure (e.g., what the takeaway is).
3. Need to cite all the explanation methods that are introduced in the paper (e.g., arch, occ).
4. The paper's argument appears to equate 'faithfulness' with 'agreement/consistency' between explainers. While the theory links common (un)faithfulness metric to interaction-driven disagreement, the author should specify the definition of 'faithfulness'.
5. The vision experiments rely on VGG16 and ResNet18, which are architectures from 2014 and 2015 which is outdated.
6. The proposed algorithm does not optimize the true interaction loss, which captures all interactions.

**Questions:**

Look at the weaknesses

---

> ### Author Response · Authors · 2025-11-20
> **Author Response**
>
> We first thank you for acknowledging the rigor of our theoretical contribution and soundness of our empirical results. We have updated the manuscript while taking care to incorporate your feedback (see text in blue).
>
> ## Weaknesses
>
> > The problem statement in the introduction is very broad. The described challenge is fundamental to almost all xai research.
>
> We agree that the problem we tackle (inconsistencies between explanations and unfaithfulness metrics) is very broad and applies to most xai research. However, we think this is a consequence of  XAI not being a mature field. Many fundamental problems remains to be tackled. Notably, the community still does not agree on what is a **good** explanation. Our work offers an interesting perspective on the problem based on functional decomposition, and a practical algorithm to address this problem.
>
> > Need to cite all the explanation methods that are introduced in the paper (e.g., arch, occ).
>
> Every explainability method used in our experiments PDP/PFI/SHAP/Arch/Occ/IG/RISE/LIME is cited when introduced in sections 2.2 and 2.3. We introduce LIME in section 2.3 (later than the others) because it is defined as the explanation that minimizes $\underline{F}$ with $w(U) = 2^{-D}$.
>
> > The readability of the figures are poor, also the main text needs to provide guidance on how to interpret the results in the figure (e.g., what the takeaway is).
>
> We thank you for this comment. Accordingly, we have updated the captions of figures 4, 5, 8, 9 in the new manuscript. The new caption describes in more detail what the figure represents and what conclusions are drawn from it.
>
> > The paper's argument appears to equate 'faithfulness' with 'agreement/consistency' between explainers. While the theory links common (un)faithfulness metric to interaction-driven disagreement, the author should specify the definition of 'faithfulness'.
>
> The definition of *(un)faithfulness* in the XAI literature is as blurry as the definition of *explanation*. There is no consensus on what is a good (un)faithfulness metric, but the general intuition is that these metrics should highlight how closely the explanation matches the true behavior of the model. Instead of coming up with the correct notion of (un)faithfulness, our paper focuses on inconsistencies between existing quality metrics for explanations, and how these inconsistencies are fundamentally related to disagreements between explanations themselves.
>
> > The vision experiments rely on VGG16 and ResNet18, which are architectures from 2014 and 2015 which is outdated.
>
> We agree with you that more recent architectures should be added to the ImageNet experiments to show how AGREED performs on more complex models. We decided to add new results using the ConvNext architecture [1] (2022) that combines ResNet and Transformers to obtain performance and scalability on par with Vision Transformers. Figure 9 now highlights results on ConvNext and the insights are similar to ResNet18. Figure 9 a) reports $L_2$ explanation disagreements for various patch sizes while (b-c-d) report the INFD metric of Arch, Occ, and IG respectively. We again see that, as pixels are grouped together by AGREED (average group size increases), disagreements and (un)faithfulness metrics are reduced. Other grouping methods (Quickshift and Square) do not exhibit similar reductions in disagreement/(un)faithfulness as they increase the size of the pixel groups.
>
>
> > The proposed algorithm does not optimize the true interaction loss, which captures all interactions.
>
> Our loss function indeed only penalizes the interactions between-groups and not within-groups. For example, let $f(x)=x\_1 x\_2 x\_3 + x\_4$ with $x\sim U([-1, 1])^4$. Our loss would be null when considering $x\_1,x\_2,x\_3$ as a single group, ignoring the interactions between these three features.
>
> Yet, feature interactions can still be recovered once groups have been identified by AGREED. Letting $x_1,x_2,x_3$ be part of a single group, their joint-PDP is
>
> $$\phi^{\text{joint-PDP}}(f, x) =  \mathbb{E}\_{b\sim\mathcal{B}}[f(x\_1, x\_2, x\_3, b\_4)] - \mathbb{E}\_{b\sim\mathcal{B}}[f(b)].$$
>
> By Definition 2.1 (Marginal Decomposition), this joint-PDP can be further decomposed as 3 main effects, 3 pairwise interactions and 1 three-way interactions.
>
> $$\phi^{\text{joint-PDP}}(f, x) = f\_{1,\mathcal{B}}(x\_1) + f\_{2,\mathcal{B}}(x\_2) + f\_{3,\mathcal{B}}(x\_3) +  f\_{12,\mathcal{B}}(x\_1, x\_2) + f\_{13,\mathcal{B}}(x\_1, x\_3) + f\_{23,\mathcal{B}}(x\_2, x\_3) + f\_{123,\mathcal{B}}(x\_1, x\_2, x\_3).$$
>
> Now, whether a practitioner would want to visualize a single three-variate function $\phi^{\text{joint-PDP}}(f, x)$ or seven functions $f_{u,\mathcal{B}}$ depends on whether the application really requires separating each feature interaction from joint effects.
>
> [1] Liu, Zhuang, et al. "A convnet for the 2020s." Proceedings of the IEEE/CVF conference on computer vision and pattern recognition. 2022.

---

### Official Review · Reviewer_5SdD · 2025-11-01

**Soundness:** 2
**Presentation:** 2
**Contribution:** 2
**Rating:** 4
**Confidence:** 3

**Summary:**

The paper aims to address the problem of XAI disagreement. The aauthors extend Functional Decomposition and parition the input features/pixels into groups and prove that the primary reason of disagreement between different explanation methods is the between-group interactions.Further, they advocate eliminating between-group interactions and show that the disagreements can be reduced by adaptively grouping features/pixel.

**Strengths:**

1) The paper deals with an important aspects of XAI: disagreement between explanation methods and the unreliability of fidelity metrics

2) The theoretical sections are formally presented, with clear conditions, properties, and proofs.

3) The proposed method was shown to work in tabular data as well as images.

**Weaknesses:**

1)  The authors use L2 disagreement between attribution vectors to define their partition loss. While this choice yields a differentiable and additive objective but it emphasizes only magnitude alignment and misses out on rank alignment. Rank-based measures such as Spearman, Kendall correlation etc could capture rank consistency between explanations but are non-differentiable for the groupwise optimization. A brief justification of this design choice would enable the readers to understand the paper clearly.

2) The partition loss formulation is conceptually appealing but it requires computing multiple attribution maps per input, severely limiting scalability. This limitation should be discussed more explicitly as it constrains the applicability of the proposed framework to high-dimensional or real-time settings.

3) The authors provide the algorithm details of AGREED in supplementary and also the computation cost of O(d^2N^2). The paper would benefit from a detailed analysis of this computation cost.

4) AGREED seems to be prohibitively expensive as the cost O(d^2N^2) increases rapidly with number of features and samples. Even for 100 features and 1000 samples the number of model  inferences would be too high. Hence, there is limited scope of appying AGREED on high dimensional and real-world settings. I request the authors to clarify this limitation in the paper.

**Questions:**

I request the authors to address the weaknesses

---

> ### Author Response · Authors · 2025-11-20
> **Author Response**
>
> We want to thank you for your insightful review that highlights the strengths of our work, but also aspects on which it can be improved.
> We have updated the manuscript to address your concerns (see the changes in blue).
>
> ## Weaknesses
> ### Other disagreement metrics
> We have modified the manuscript to better motivate the use of the $L_2$ disagreement metric. Firstly, section 2.3 has been extended by citing the work of [1] who used the pearson/spearman correlation is explanation agreement metrics. Section 2.3 now briefly hints that the $L_2$ disagreement is chosen because it respects theoretical properties that the other do may not.
>
> Second, after presenting theorem 3.3 in section 3, we now accentuate the choice of $L_2$ disagreement over correlation metrics which might break properties 2 & 3. Breaking property 2 is especially problematic, since increasing agreement no longer guarantees that we have fused two groups that interact.
>
> In fact, we can easily come up with toy examples where correlation breaks property 2. Let us use the following metric slightly modified from [1]
> $$\text{Agreement}(\phi^1, \phi^2) = \frac{\langle \phi^{1}, \phi^2\rangle}{\\|\phi^1\\| \\|\phi^2\\|}$$
> to measure the cosine similarity (equivalent to the Peason correlation up to a translation of the vectors $\phi$). Let $x\in \mathbb{R}^4$ be 4 features fed to a model that is additive w.r.t features 1 and 2 : $f(x)=f_1(x_1) + f_2(x_2) + f_{34}(x_3,x_4)$.
> Let the feature attributions be $\phi^1(f, x)=[2, 2, 0, 1]$ and $\phi^2(f, x)=[2, 2, 1, 0]$ according to two methods (note that the attributions of $x_1$ and $x_2$ agree between both methods since these features do not interact).  The cosine agreement is $\frac{8}{9}$ when no features are grouped. Yet, if we group features 1 and 2,
> the group attributions become $\phi^1(f, x,\mathcal{P})=[4, 0, 1]$ and $\phi^2(f, x,\mathcal{P})=[4, 1, 0]$ and the agreement is now $\frac{16}{17}$. Thus, we can increase agreement (reduced disagreement) without creating a meaningful feature group: the model was already additive w.r.t features 1 and 2.
>
> ## Scalability
>
> Weaknesses 2, 3 and 4 are all related to the topic of scalability so we address them jointly. We have updated the last paragraph of section 3 to better accentuate the complexity $\mathcal{O}(d^2 N^2)$ and how we handle it for experiments on tabular and images.
>
> ### Tabular data
> For tabular data, we have to differentiate the experiments where we report (un)faithfulness metrics at each step of the AGREED algorithm (Figures 4 & 5) from the experiments where we run AGREED once and report qualitative results (appendix E.2.2).
> Figures 4 & 5 required subsampling $50\leq N\leq 100$ data samples to ensure reasonable runtimes because of the scale of the
> experiments. For five different seeds, we must run AGREED and each time we fuse feature groups we must compute SHAP explanations and the Sensitivity-1/INFD/SWF (un)faithfulness metrics of PDP/SHAP/PFI (9 combinations). While these experiments highlight the effectiveness of feature grouping for reducing explanation disagreements and (un)faithfulness, it is not representative of how AGREED would be used by practitioners.
>
> In practice, we run AGREED on tabular data only once and stop once the disagreement between PDP and PFI falls below a threshold. We do not need to log the various metrics of Figures 4 & 5, which allows us to increase the number of samples to $N=1000$ while having a runtime of at most a minute (see the updated Appendix E.2.2).
>
> ### Images
> On images, we use $N=2$ (the image and the baseline) so the bottleneck is the $d^2$ factor, which we handle by first grouping the pixels into $W\times W$ patches that we subsequently fuse. This process must be done on each image separately, which prohibits real-time application. In section 5.2.2 we clarify that the runtime of AGREED takes about 1-7 seconds per image on VGG16/ResNet18 and 4-30 seconds for ConvNext. While not instantaneous, we argue that AGREED could still be used to interpret the model on failure cases, of which there may be hundreds. Obtaining saliency maps on 100 images would take about less than an hour.
>
> [1] Schwarzschild, Avi, et al. "Reckoning with the disagreement problem: Explanation consensus as a training objective." Proceedings of the 2023 AAAI/ACM Conference on AI, Ethics, and Society. 2023.

---

### Official Review · Reviewer_MwSX · 2025-11-01

**Soundness:** 3
**Presentation:** 3
**Contribution:** 2
**Rating:** 6
**Confidence:** 4

**Summary:**

Summary
This paper addresses the disagreement problem among post-hoc explanations in explainable AI by proposing an adaptive feature grouping method (AGREED) that reduces interactions between feature groups, aiming to bring explanation methods into agreement. The work builds on prior frameworks of functional decomposition and attribution methods, extending them to disjoint feature groupings. Empirical validation on tabular and image datasets shows improvements in explanation consistency and faithfulness metrics.

Soundness
The methodology is mathematically sound and supported by formal proofs linking disagreement to between-group interactions. The algorithm is intuitive and empirically demonstrated to reduce disagreement. Results are robust on multiple datasets, though assumptions like groupwise additivity in regions and sampling baselines could affect results in practice.

Presentation
The paper is well-organized but the presentation is somewhat standard and occasionally dense. Some of the core ideas could be clarified with more intuitive or visual illustrations, especially around interpretation of grouped attributions. The empirical section is comprehensive yet could better emphasize practical significance. Comparisons to prior grouping methods on images are limited by scalability and implementation constraints, leading to incomplete benchmarking in some cases.

Literature review
Literature review is a bit outdated, check these e.g. for attributions methods such as IG and SHAP with stronger mathematical content:

A General Feature Attribution Framework under a Black-box Setting

-Y. Cai, A. Thibaud, G.Wunder, International Conference on Machine Learning (ICML'25), Vancouver, Canada, 2025

-On Gradient-like Explanation under a Black-box Setting: When Black-box Explanations Become as Good as White-box
Y. Cai, G. Wunder, International Conference on Machine Learning (ICML'24)

Contribution
-Proposes a practical adaptive grouping algorithm to reduce disagreement in feature attributions
-Unifies explanation methods under a functional decomposition framework for groups
-Extends interpretability to handle feature group interactions explicitly
-Demonstrates empirical gains on tabular and image data
-Provides theoretical insights into when explanation agreements are possible

**Strengths:**

-Sound theoretical grounding
-Addresses a key challenge limiting trust in explainability
-Novel focus on attributing disagreement to feature interaction

**Weaknesses:**

-Novelty incremental given prior functional decomposition frameworks; core insight of grouping to minimize interactions is expected
-Presentation could benefit from more accessible examples and clearer motivation for practitioner
-Limited direct comparisons with some existing grouping algorithms on images
-Interpretation of grouped attributions remains difficult without actionable visualization
-Image domain grouping requires per-sample runs, limiting real-world scalability

**Questions:**

-How does AGREED compare to prior grouping methods that are not feasible for images, when run on smaller image subsets?
-How practically interpretable are grouped attributions for non-expert users, especially with large feature clusters?
-Can the algorithm support global grouping for an entire dataset rather than per-sample in images?
-Would integrating semantic/region-based features improve grouping and interpretability further?

---

> ### Author Response · Authors · 2025-11-20
> **Author Response**
>
> We wish to thank you for your positive comments on the soundness of the formal results, and for sharing more up-to-date references on XAI techniques. We have updated the manuscript while taking care in incorporating your concerns (see the changes in blue).
>
> ## Weaknesses
>
> > Literature review is a bit outdated
>
> While the cited methods are not recent (2001-2020), they remain established methods implemented in source software : SHAP, Scikit-Learn, Captum, Xplique. Still, we acknowledge that improvements to these methods are still being done to this day. For instance, [1] (2024) adapts the IG to a truly black-box (input-output) setting instead of requiring access to gradients. We have updated section 2.2 to highlight these recent improvements. We thank you again for sharing this recent contribution.
>
> > Novelty incremental given prior functional decomposition frameworks; core insight of grouping to minimize interactions is expected
>
> The implications of Theorem 3.3 are surprising given the existing literature. This is because only certain disagreement metrics ($L_2$ distance), and pairing between explanations (Arch and Occlusion) lead to an objective whose minimization respects three desirable properties. Most notably, the disagreements between Lime-vs-SHAP pairing cannot be considered because it might break Property 3 of definition 3.1), which is unexpected.
>
> > Presentation could benefit from more accessible examples and clearer motivation for practitioner
>
> > Interpretation of grouped attributions remains difficult without actionable visualization
>
> We agree that the main body of the paper is theoretically heavy and has less room
> for qualitative analyses to highlight how practitioners can gain insights from our explanations. Due to space constraints, we were forced to leave such qualitative analyses in appendix E.2.2 (tabular) and E.4 (images). These sections highlight how to interpret feature groups. For Tabular data, as long as the groups contains no more than 3 features, it is possible to plot the group attribution (see Figures 16 and 18). Visualizing larger groups size on tabular data is a challenge that we are currently not able to address.
> On images, groups are visualized as patches overlaid over the image. We argue that these patches are interpretable (at least more interpretable than individual pixels) since they highlight large regions that the model is using to predict.
>
> > Limited direct comparisons with some existing grouping algorithms on images
>
> We address this point along-side **Question 1** below.
>
> > Image domain grouping requires per-sample runs, limiting real-world scalability
>
> Section 5.2.2 clarifies that AGREED takes 1-30 seconds per image. While explanations are not real-time, explanations for 100 instances (possibly hand-picked failure cases of the model), can be computed in less than an hour.
>
> ## Questions
>
> > How does AGREED compare to prior grouping methods that are not feasible for images, when run on smaller image subsets?
>
> Other grouping methods (Recurse, iGreedy, Pairwise) are not applicable to imagenet for many reasons besides the size of the images. Pairwise is not applicable because the patches it generates are not guaranteed to be path-connected i.e. disjoint patches over the image could be considered a single "group". Pairwise only yielded patch-connected patches on the toy image dataset because the model was group-wise additive by design.
>
> Recurse and iGreedy were developed for tabular data and have no notion of pixel vicinity. If we were to reimplement them for Images, they would still be unable to generate path-connected patches on ImageNet.
>
> > How practically interpretable are grouped attributions for non-expert users, especially with large feature clusters?
>
> See comment above on interpreting feature groups and actionable visualization.
>
> > Can the algorithm support global grouping for an entire dataset rather than per-sample in images?
>
> This is possible in theory, but we expect the algorithm to yield a single patch that covers all pixels. Running the algorithm separately on each image already leads to large patches (see Figures 30 & 31 in the new manuscript).
>
> > Would integrating semantic/region-based features improve grouping and interpretability further?
>
> As highlighted in our discussion with **reviewer EP8L (Question 3)**, we think that applying AGREED on the function that maps pixels to hidden layers can help the grouping algorithm (allow for overlapping between patches) as well as making the explanations more semantically meaningful. This is part of our future work.
>
> [1] On Gradient-like Explanation under a Black-box Setting: When Black-box Explanations Become as Good as White-box Y. Cai, G. Wunder, International Conference on Machine Learning (ICML'24)

---

### Official Review · Reviewer_EP8L · 2025-11-01

**Soundness:** 4
**Presentation:** 3
**Contribution:** 3
**Rating:** 6
**Confidence:** 4

**Summary:**

This paper addresses the issue that many explanation methods often disagree with each other. They formally prove that when the model is group-wise additive, using the ground truth groupings, then unfaithfulness metrics will all reduce to 0. They then prove that if we minimize the difference between Arch and Occlusion, then this will minimize the disagreement between all groups. And they achieve it by iteratively merging groups with the highest pairwise interactions. They repeat until the disagreement falls below a threshold. They conduct experiments to show that when they merge the groups progressively, the difference between different explanation methods consistently decreases, and that each method is changing to become more agreed as the loss decreases, and that the proposed method AGREED can get much lower disagreement when merging not many groups than other methods, meaning that it is merging the groups that will reduce disagreement the most.

**Strengths:**

1. The paper proves a novel and important theorem that if we reduce the disagreement between Arch and Occlusion, it will reduce the disagreement for all methods.
2. Their proposed method empirically finds better groups than other methods.
3. The authors conducted comprehensive experiments to demonstrate their advantage in different aspects.

**Weaknesses:**

1. Some of the empirical figures (e.g., Figures 4 and 5) are not very intuitive
in their current presentation, and the takeaways are not made explicit in the
captions. Clarifying the intended interpretation in the captions or main text
would significantly improve readability for non-expert readers.

2. One prior work also proves how one can achieve zero unfaithfulness when using the correct grouping structure. It would be helpful to cite this work to situate the current contribution within a more comprehensive body of related theory [1].

3. The paper considers only non-overlapping partitions, while there could be cases where groups overlap with each other a lot and only considering non-overlapping groups could make it less expressive.

[1] You et al. "Sum-of-Parts: Self-Attributing Neural Networks with End-to-End Learning of Feature Groups" ICML 2025

**Questions:**

1. You et al. 2025 also considers group-based attributions and categorizes insertion and deletion style errors (Definition 2.2 of Sum-of-Parts is similar to Equation (10) in this paper, and Definition A.1 is similar to Equation (9)), although they compute differences with respect to the prediction contributed by a group, rather than differences in attribution scores produced by two methods for that group. They show that when there are between-group interactions (e.g., polynomial correlations), these errors can grow exponentially (Theorem 2.3), while using the correct grouping drives the error to zero (Theorem 2.4). The theoretical results here (e.g., Theorem 3.2) appear closely related, as both hinge on the vanishing of between-group interaction terms. It would be helpful for the authors to explicitly discuss the similarities and differences with You et al. 2025, including the choice of disjoint partitions in this paper vs. allowing overlapping groups in You et al 2025.

2. In Section 5.2.1 and Figure 7, only Arch and Occlusion are shown. Since the paper's claims concern reduced disagreement across a broad set of methods, could you clarify whether the qualitative improvements also generalize to other methods like LIME/SHAP etc.? Including additional visual examples in the appendix would strengthen the qualitative evidence.

3. In functions with overlapping interaction structure (e.g., where a feature participates in multiple strongly interacting cliques), disjoint partitions may not be expressive enough. In such cases, the optimization may be forced to merge many features together, reducing interpretability. It would be helpful if the authors could comment on this tradeoff and whether and how overlapping or hierarchical groupings could be supported in future work.

[1] You et al. "Sum-of-Parts: Self-Attributing Neural Networks with End-to-End Learning of Feature Groups" ICML 2025

---

> ### Author Response · Authors · 2025-11-20
> **Authors Response**
>
> We would first like to thank you for acknowledging the important theoretical contributions underlying AGREED and for your insightful feedback, especially regarding the existing literature on overlapping pixel patches. We have updated the manuscript while taking care of incorporating your comments (new content is in blue).
>
> ## Weaknesses
>
> > Some of the empirical figures (e.g., Figures 4 and 5) are not very intuitive in their current presentation, and the takeaways are not made explicit in the captions. Clarifying the intended interpretation in the captions or main text would significantly improve readability for non-expert readers.
>
> We have updated the captions of Figures 4, 5, 8, 9. The new caption both describes what the Figure represents, and what is its main takeaway.
>
> > One prior work also proves how one can achieve zero unfaithfulness when using the correct grouping structure. It would be helpful to cite this work to situate the current contribution within a more comprehensive body of related theory [1].
>
> We have extended the section 2.3 that introduces unfaithfulness metrics. The section now cites [1] and clarifies that alternative work considers overlapping feature groups. Furthermore, we also refer to a new appendix B.3.2 that discusses the overlap-based unfaithfulness metrics in more details. More on this appendix below.
>
> > The paper considers only non-overlapping partitions, while there could be cases where groups overlap with each other a lot and only considering non-overlapping groups could make it less expressive.
>
> In Section 4 (Related Work), we have extended the discussion on overlapping vs non-overlapping groups. Notably, we added "" AGREED might be forced to merge many features together, hindering interpretability compared to model-specific methods that allow for overlap."". The manuscript is now more transparent regarding this weakness of the method.
>
> ## Questions
>
> ### Question 1
> We thank you for sharing this work on overlapping feature groups. The InsErr and DelErr proposed in [1] are in fact generalization of the $\underline{F}$ and $\overline{F}$ metrics used in our paper. When groups happen to form a partition, then InsErr falls back to $\underline{F}$ and DelErr falls back to $\overline{F}$. We added a new appendix B.3.2 to discuss the differences between said metrics. Most notably, overlap and non-overlap metrics are null for different types of models. Unfaithfulness metrics without overlap are null whenever the model is group-wise additive, which is the main motivation behind AGREED. Unfaithfulness metrics with overlap are null when the model has a specific polynomial form (Equation 21 in the new manuscript). We present a counter-example with a trigonometric model that does not respect Equation 21 and show that InsErr is no longer null even with the correct groups.Consequently, we think that the choice of overlap vs no-overlap in unfaithfulness metrics (and explanation) is dependent on the assumptions we are allowed to make on the model ($f$ is groupwise additive vs $f$ respects Equation 21).
>
> ## Question 2
>
> We have added Appendix E.3 to show that agreement is also reached for other explainability methods on the toy image data.
> This appendix is now referenced in the main manuscript at Section 5.2.1.
>
> ## Question 3
>
> The last paragraph of section 5.2.1 (discussing saliency maps on ImageNet) has been updated to accentuate the practical limitations of requiring a feature partition (no overlap). This constraint in fact explains why our results on MiniImageNet tend to generate a single patch that covers the whole object. As future work, we think that applying AGREED to explain the pixel-neuron mapping (instead of the pixel-logit), will help generate overlapping patches.
>
> For example, let $h:\mathbb{R}^d\rightarrow \mathbb{R}\_+^C$ be a feature extractor that maps images to $C$ channel activations (post ReLU). The final output is the composition of these activations with a linear layer $f(x) = \sum_{c=1}^C \omega_c h_c(x)$. Unlike Self-Attributing NNs, each function $h_c$ potentially depends on all pixels $x$. Yet, in the region $R=\prod_{i=1}^d[0, x_i]$ covering an image and the baseline $b=0$, each map $h_c$ might be groupwise-additive. This would occur if disjoint regions of the image are responsible for the difference $h_c(x) - h_c(0)$ on this single channel. Applying AGREED on different channels $h_c$ would highlight these regions, while allowing for overlapping between the patches derived from different channels.

---

> > ### Comment · Reviewer_EP8L · 2025-11-28
> >
> > Thank you very much for the detailed responses and update of the manuscript. The discussion and the example to show how the unfaithfulness score is no longer 0 when the functions are approximately group-wise additive instead of polynomial is very interesting! I would like to raise the rating to 8, but somehow I cannot find the edit button anymore.

---

### Author Response · Authors · 2025-11-20
**Official Comments by the Author**

Dear reviewers,

we would like to thank all of you for taking the time to review our work, for highlighting the quality of our contributions, and for proposing means of improving the paper. Taking your feedback into account, we have updated the manuscript and highlighted all changes in blue.

While we respond to your questions directly as comments on openreview, we encourage you take a look at the manuscript to see the full changes. Any change in the manuscript that you need to look at is referenced in our response.

---

### Meta-Review · Area_Chair_Zy9a · 2026-01-18

**Summary:**

The paper is interested in discrepancies in AI explainability metrics. They propose that, interpreting different explanation methods as functional decompositions, their discrepancy is due to interactions between groups of features. They show that grouping features can be used to reduce discrepancy. All reviewers praised the formal contributions of the paper, and the interest in the new method.

The primary concerns were questions that were largely addressed, and scalability.

**Reviewer Concerns:**

The primary outstanding concern is the large computational cost of the method. However the authors clarified clear use cases.

**Reviewer Scores:**

Reviewers EP8L and MwSX were supportive and had their questions answered, so they are likely to keep their scores. Reviewer Dpg6 was borderline and had many of their questions answered, and one of their concerns addressed by a new experiment; they are likely to go to a 6. Reviewer 5SdD received clarification, but did not fully have their concern about scalability addressed, so they may have remained at a 4 or gone up to a 6.

---

### Decision · Program_Chairs · 2026-01-26

Accept (Poster)